# Evidence of a distinct collective mode in Kagome superconductors

Bin Hu [1,2,5], Hui Chen [1,2,3,5], Yuhan Ye[1,2,5], Zihao Huang[1,2], Xianghe Han[1,2], Zhen Zhao[1,2], Hongqin Xiao[1,2], Xiao Lin[1,2], Haitao Yang [1,2], Ziqiang Wang [4] ✉ & Hong-Jun Gao [1,2,3] ✉

The collective modes of the superconducting order parameter fluctuation can provide key insights into the nature of the superconductor. Recently, a family of superconductors has emerged in non-magnetic kagome materials $AV_3Sb_5$ ($A$ = K, Rb, Cs), exhibiting fertile emergent phenomenology. However, the collective behaviors of Cooper pairs have not been studied. Here, we report a distinct collective mode in $CsV_{3-x}Ta_xSb_5$ using scanning tunneling microscope/ spectroscopy. The spectral line-shape is well-described by one isotropic and one anisotropic superconducting gap, and a bosonic mode due to electron-mode coupling. With increasing $x$, the two gaps move closer in energy, merge into two isotropic gaps of equal amplitude, and then increase synchronously. The mode energy decreases monotonically to well below $2\Delta$ and survives even after the charge density wave order is suppressed. We propose the inter-pretation of this collective mode as Leggett mode between different super-conducting components or the Bardasis-Schrieffer mode due to a subleading superconducting component.

The collective bosonic modes, which usually manifest as a peak-dip-hump feature in the differential tunneling conductance spectrum, can provide critical information for understanding a wide range of super-conductors. The typical example is the phonon mode and electron-phonon coupling in conventional superconductor Pb and unconven-tional high-$T_c$ cuprate superconductor $Bi_2Sr_2CaCu_2O_{8+\delta}$ observed by scanning tunneling microscope/spectroscopy (STM/S)[1,2]. Candidate magnetic resonance modes in electron doped high-$T_c$ cuprate $Pr_{0.88}LaCe_{0.12}CuO_4$[3] and one-unit-cell iron-based superconductor FeSe[4] have also been observed by the same method. In a variety of uncon-ventional superconductors, a linear relation between the collective mode energy and the superconducting (SC) transition temperature $T_c$ has been revealed by the well-known Uemura plot[5]. Collective modes can also originate directly from the SC order parameter fluctuations[6–12]. These SC modes, the amplitude or the Higgs mode[6,7], the Nambu–Goldstone phase mode[7,13], and the relative phase or the Leggett mode[12,14] in multi-

component superconductors, etc. can offer insight into the nature of the pairing state[6,7,15]. Detecting the SC modes has been a real challenge and only recently progress has been made in this direction[7–10,14,16–22], such as the Leggett mode has been observed in $MgB_2$[19–21], Fe-based superconductors[22], and monolayer $NbSe_2$[14].

Recently, transition-metal kagome lattice materials have been shown to exhibit many intriguing properties in the metallic phase impacted by geometric frustration, flat bands, Dirac fermion band crossings, and van Hove singularities, making them a fertile ground to study the interplays among electronic geometry, topology, and correlation[23–25]. The field leaps forward with the discovery of vanadium-based nonmagnetic kagome superconductors $AV_3Sb_5$ ($A$ = K, Rb, and Cs)[26–30]. In the normal state, $AV_3Sb_5$ shows intriguing rotational symmetry-breaking[31–34] and potentially time-reversal sym-metry breaking charge density wave (CDW) order[35–39]. Below $T_c$, the SC state coexists with CDW order and exhibits strong coupling

[1]Beijing National Center for Condensed Matter Physics and Institute of Physics, Chinese Academy of Sciences, 100190 Beijing, PR China. [2]School of Physical Sciences, University of Chinese Academy of Sciences, 100190 Beijing, PR China. [3]Hefei National Laboratory, 230088 Hefei, Anhui, PR China. [4]Department of Physics, Boston College, Chestnut Hill, MA 02467, USA. [5]These authors contributed equally: Bin Hu, Hui Chen, Yuhan Ye. ✉e-mail: wangzi@bc.edu; hjgao@iphy.ac.cn

superconductivity[40], multiple SC gaps[41–44], and gap anisotropy[42–44]. In addition, evidence for pair density wave (PDW) formation, pseudogap pheonmena[40,45], chiral superconductivity[46], and change-$6e$ ($4e$) flux quantization[45,47], provides a new materials platform for studying fundamental and unsolved issues that also arise in other superconductors such as the cuprate[40]. However, the nature of the kagome superconducting state, whether the superconductivity is multiband, whether the superconducting collective mode exists, and how the mode and superconductivity coevolve with the CDW/doping remain unexplored in kagome superconductors.

Here, we study the properties of the SC state and collective excitations in $CsV_3Sb_5$ as well as their evolution with the weakening and vanishing of the CDW order in Ta-substituted $CsV_{3-x}Ta_xSb_5$ by utilizing ultra-high energy resolution scanning tunneling microscope/spectroscopy. In the crystalline structure of $CsV_3Sb_5$, V atoms form the kagome lattice (inset of Fig. 1a), where the CDW and superconductivity coexist below $T_c$. With Ta-substitution, the CDW transition temperature decreases while the SC $T_c$ increases, the CDW disappears when

the substitution ratio $x$ is bigger than 0.3 (Fig. 1a). We observe prominent peak-dip-hump structures in the tunneling conductance spectra which we attribute to the renormalized single-particle density of states in the presence of electron-mode coupling[2–4,14,48,49]. Two SC gaps can be extracted from the conductance line-shape, one isotropic and one anisotropic, with different pair-breaking strengths. With increasing $x$, the anisotropic gap evolves into an isotropic one. We identify a single collective mode whose energy reduces from just below the single-particle excitation gap $2\Delta$ at small $x$, to well-below $2\Delta$ at large $x$. The collective mode survives when CDW is fully suppressed. We discuss the nature of the bosonic collective mode and suggest that our findings are most consistent with the relative phase mode, i.e. the Leggett mode in the multi-band kagome superconductors, which may couple to the amplitude mode as a Higgs-Leggett mode.

## Results

$CsV_3Sb_5$ has the highest SC transition $T_C \sim 2.5\,K$ among the $AV_3Sb_5$ family[29]. The weak bonding between the Cs layer and the Sb layer leads

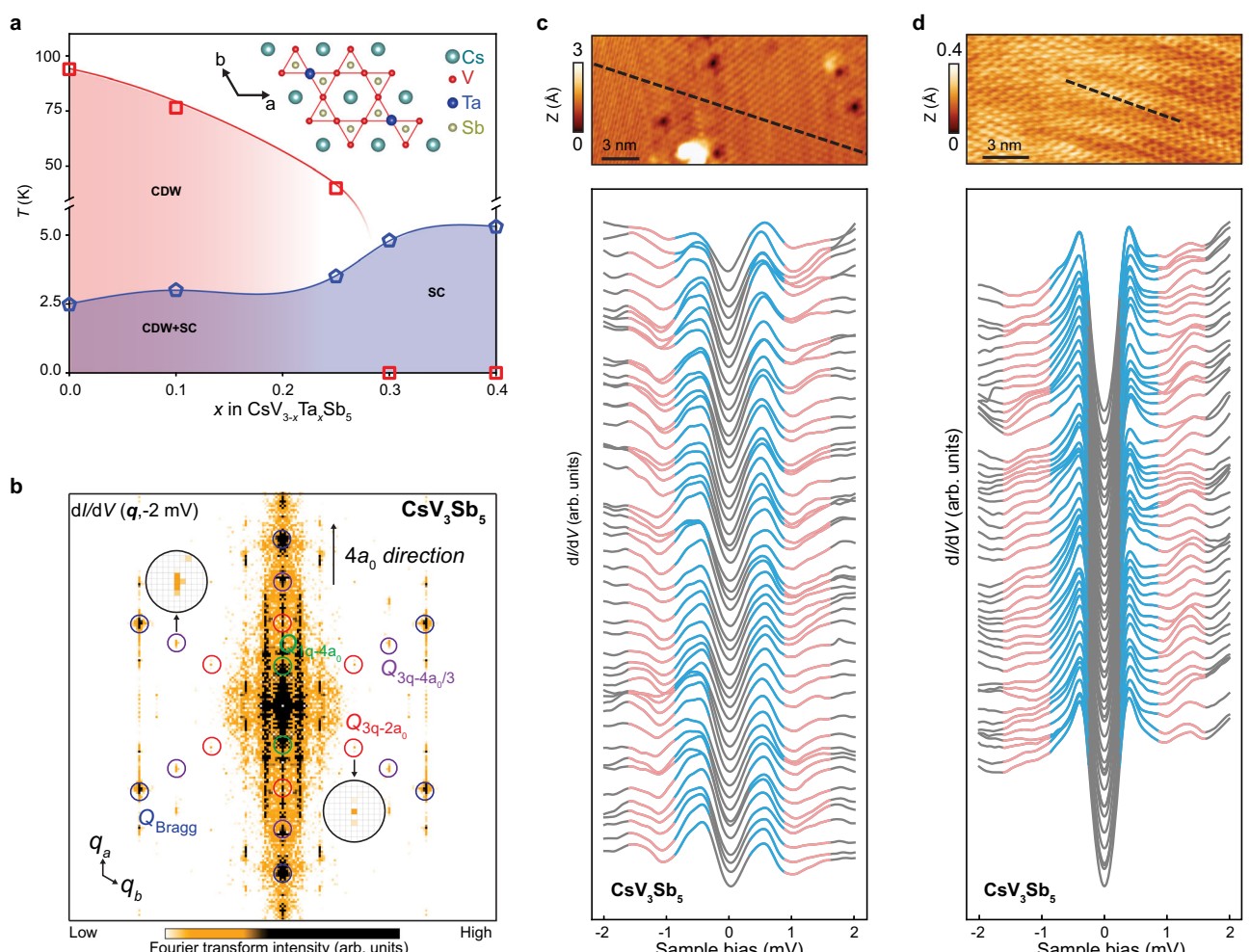

**Fig. 1 | Observation of bosonic mode in $CsV_3Sb_5$. a** Phase diagram of $CsV_{3-x}Ta_xSb_5$. The superconducting (SC) transition temperature increases while the charge density wave (CDW) transition temperature decreases with increasing $x$. The CDW is undetectable when $x > 0.30$. The transition temperatures of SC and CDW are obtained from transport measurement. The inset of (**a**) shows the crystal structure of $CsV_3Sb_5$. **b** The Fourier transform of the tunneling conductance d$I$/d$V$ map taken at −2mV on a typical as-cleaved Sb surface of $CsV_3Sb_5$, showing the $4a_0$ CDW, $2a_0 \times 2a_0$ CDW, $4a_0/3 \times 4a_0/3$ CDW and $4a_0/3 \times 4a_0/3$ pair density wave (PDW), and intriguing quasi-particle interference patterns, respectively. The $a_0$ in (**b**) is the lattice constant. The $Q_{1q\text{-}4a0}$, $(Q_{3q\text{-}4a0/3})$ and $Q_{Bragg}$ in (**b**) are the wave vectors of $4a_0$ CDW, $4a_0/3$ CDW/PDW, and Bragg peaks, which are indicated by the colored circles. **c** A series of d$I$/d$V$

spectra (lower panel) obtained along the line-cut (black dotted line) on the topography in the upper panel, showing two SC gaps accompanied by a pair of peak-dip-hump features just outside the SC gaps, indicative of a bosonic mode. **d** Same as in (**c**) but over a line-cut in a different spatial region (upper panel), where the d$I$/d$V$ spectra (lower panel) are dominated by the spectral line-shape of a smaller gap in a more subtle two-gap structure. The peak-dip-hump features outside of the SC gaps are also clearly visible indicating the presence of a bosonic mode. For (**c**, **d**), the scanning tunneling microscope (STM) scanning parameters are bias setup ($V_s$) = −600 mV, current setup ($I_t$) = 400 pA and $V_s$ = −20 mV, $I_t$ = 1 nA, respectively, and the scanning tunneling spectroscopy (STS) parameters are the same: $V_s$ = −2.0 mV, $I_t$ = 1 nA, and lockin modulation amplitude ($V_{mod}$) = 0.05 mV.

to Sb and Cs cleaved surfaces. Large-scale and clean Sb surfaces closest to the V kagome layer are obtained through the STM tip "sweeping" process[40]. The cascade of symmetry-breaking electronic density wave orders discovered on the Sb-terminated surface of CsV$_3$Sb$_5$ is visible in the Fourier transform of the low-energy d$I$/d$V$ maps at low-temperature on top of the quasi-particle interference pattern, including the 3Q-CDW with $2a_0 \times 2a_0$ periodicity, the unidirectional $4a_0$ charge-stripes, and the 3Q-PDW with $4a_0/3 \times 4a_0/3$ spatial modulations (Fig. 1b).

We first perform detailed STS studies in the SC state of CsV$_3$Sb$_5$ at a base temperature of 50 mK, which is much lower than $T_C$. We observe two different Sb surface regions, where the d$I$/d$V$ spectra either show directly two SC gaps or a single SC gap. The d$I$/d$V$ spectra are shown with two SC gaps (Fig. 1c) and one dominant SC gap (Fig. 1d) along two line-cuts indicated on the topographies in the two regions, respectively. Imaging of the two-gap structure is known to be sensitive to the STM tip condition[41]. The possible reasons could be that the two SC gaps stem from different bands, similar to the observation in iron-base superconductor FeSe[50], or carry different form factors that modify the tunneling matrix elements.

Strikingly, just outside the symmetric SC coherence peaks, a pair of peak-dip-hump structures (highlighted by the red color in Fig. 1c, d) are clearly observed. These peak-dip-hump structures are usually linked to the collective excitations, arise from the renormalization of electron self-energy[51] or inelastic tunneling processes[52]. To identify the SC gaps and the energy of bosonic mode, we plot two typical d$I$/d$V$ spectra taken from the two-gap and the dominant one-gap regions respectively in Fig. 2a, and the corresponding d$^2$$I$/d$V^2$ spectra (Fig. 2b) that allow a more quantitative determination of the energies of the peak-dip-hump features. The two SC gaps $\Delta_{s,b}$ are easily extracted from the coherence peak-to-peak distance in the blue two-gap spectrum. While the small gap $\Delta_s$ matches well with the peak-to-peak distance in the red one-gap spectrum, indicating the smaller SC gaps in the two regions are identical, the large gap $\Delta_b$ is not obvious in the one-gap spectrum dominated by the small gap, but rather hides in and is responsible for the elevated tails of the small gap coherence peaks (Fig. 2a). Due to the presence of two SC gaps, the bosonic mode with energy $\Omega$ is pushed outside the single-particle gaps and shows up as dip-hump structures in the d$I$/d$V$ curves, which are determined by the dips or peaks in the d$^2$$I$/d$V^2$ spectra at energies $E_{s,b}^{\pm} = \pm(\Delta_{s,b} + \Omega)$ (Fig. 2b). We choose the most prominent and symmetric local minima/maxima that is closest to the coherence peak, while the other peaks/dips may be the data noise due to the weakness and particle-hole asymmetry. This allows us to determine the mode energy $\Omega$.

We determine the gap sizes and mode energies (details in "Methods") from 300 d$I$/d$V$ spectra obtained in various regions and on different samples including the cases of Fig. 1c and d and plot them as histograms in Fig. 2c. The statistical average values are $\Delta_s = 0.40(0.09)$ meV, $\Delta_b = 0.62(0.09)$ meV, $|E_s| = 1.16(0.09)$ meV, and $|E_b| = 1.42(0.09)$ meV. The average energy of the bosonic mode is obtained from $\Omega = (|E_{s/b}^+| + |E_{s/b}^-| - 2\Delta_{s/b})/2$ to be 0.76(0.13) meV offset by the small gap and 0.80(0.13) meV by the big gap, which implies a single collective mode within the energy resolution in the superconductor in spite of having two SC gaps (Fig. 2c).

In Fig. 2d, we show a series of d$I$/d$V$ curves with the dominant one-gap line-shape in a stack-plot without offset from different regions and samples. The spatial distribution of $\Delta(r)$ and $\Omega(r)$ within a field of view of 15 nm × 15 nm are presented in Supplementary Fig. 2. It highlights the spatial nonuniformity of the superconductor with significant variations of the coherence peaks, in-gap conductance, and the Bogoliubov quasiparticle density of states beyond the SC gap. The spatially-averaged d$I$/d$V$ spectrum is shown as the red curve in Fig. 2d. The dip-peak-hump features are clearly visible in the averaged conductance, indicating the robustness of the bosonic mode against nonuniformity. Taking the derivative of the averaged conductance, i.e. the d$^2$$I$/d$V^2$

curve in Fig. 2e, the averaged SC gap and the bosonic mode energy can be determined, which are consistent with the statistical analysis of individual spectrum shown in Fig. 2c. It should be noted that the determination of mode energy in d$^2$$I$/d$V^2$ does not directly imply an inelastic mode, and the difference between elastic and inelastic methodology falls into the error bars.

We attempt at a description of the averaged differential conductance line-shape using the Dynes functions. The dominant small gap coherence peaks and the V-shape in gap density of states suggest that the small gap is anisotropic. Moreover, the analysis in Fig. 2a-b suggests that a large gap is present and must be included to account for the elevated flanks of the coherence peaks. Thus, we use the two-gap Dynes formula to describe the spatially-averaged d$I$/d$V$ spectrum in Fig. 2f, one anisotropic gap and one isotropic gap, as detailed in "Methods" sections. We note this SC gap structure is consistent with previous studies of superconductivity in CsV$_3$Sb$_5$[41,43,53]. Specifically, we use an anisotropic gap function $\Delta_1 = \Delta_{1,\max}\cos^2 3\theta + \Delta_{1,\min}\sin^2 3\theta$ that varies in the range $[\Delta_{1,\min}, \Delta_{1,\max}]$ in the first Dynes function $D_1(E)$ with a pair-breaking parameter $\Gamma_1$, and an isotropic gap $\Delta_2$ in the second Dynes function $D_2(E)$ with a pair-breaking parameter $\Gamma_2$ ("Methods"). We find that an excellent description of the averaged d$I$/d$V$ line-shape can be achieved with $\Delta_{1,\max} = 0.353$ meV, $\Delta_{1,\min} = 0.177$, $\Gamma_1 = 0.012$ meV, $\Delta_2 = 0.629$ meV, and $\Gamma_2 = 3 \times 10^{-4}$ meV for the SC state in CsV$_3$Sb$_5$ (Fig. 2f). The values of $\Delta_{1,\max}$ and $\Delta_2$ are consistent with the statistical values of $\Delta_s$ and $\Delta_b$ in Fig. 2c, respectively. The determined gap functions are depicted on the polar plots in the inset of Fig. 2f by the red and blue lines.

To have a deeper inspection of the relation between this bosonic mode and superconductivity, we collect a series of d$I$/d$V$ spectra under different temperatures, as shown in Supplementary Fig. 3. The bosonic mode, which is manifested as the dip-peak-hump feature disappears concomitantly with the closing of superconducting gap when $T > 2.06$ K. Moreover, the dip-peak-hump structure becomes more significant when using the superconducting STM tip ("Methods"). The signature of Josephson current, superconducting coherence peaks, and the feature of the bosonic mode weaken with increasing temperature, then disappear simultaneously when $T > 2.00$ K. The complementary tunneling spectra using the normal and superconducting STM tips demonstrate the observed mode is a superconducting collective mode in the kagome superconductor. For the CsV$_3$Sb$_5$−insulator−CsV$_3$Sb$_5$ nanoflake tunneling junction, the "intra-band" Josephson tunneling between same band and the "inter-band" Josephson tunneling between different bands are all allowed. In addition, the bosonic mode manifesting itself at energy around $\Omega$ is also observed when using the superconducting STM tip ("Methods").

A few remarks are in order. First, the magnitudes of the pair-breaking fields $\Gamma_1$ and $\Gamma_2$ are different, which may suggest that the two gap functions are likely to have originated from different parts of momentum space, possibly from different bands. Physically, it is reasonable to expect that the anisotropic gap is associated with CDW reconstructed bands of the more correlated V $d$-orbitals around the zone boundary, whereas the isotropic gap with the band of the Sb $p$-orbitals around the zone center is not sensitive to CDW order[41,43,54]. Second, we note that a constant background is included to account for the zero-bias density of states in the weighted two-gap Dynes functions analysis (Methods). The origin of these residual density of states is currently unknown. Finally, in Fig. 2f, the difference between the line-shape of the Dynes description and the experimental data demarcates the bosonic mode contribution highlighted by the shaded regions.

To reveal the interplay between CDW order and superconductivity, we next study the evolution of differential conductance spectra and the collective mode in Ta-substituted single crystal CsV$_{3-x}$Ta$_x$Sb$_5$. As shown in Fig. 1a, Ta substitution suppresses long-range CDW order and enhances superconductivity[55]. The typical STM topographies on samples with $x = 0.10$, 0.25 and 0.40 are shown in Fig. 3a−c, respectively.

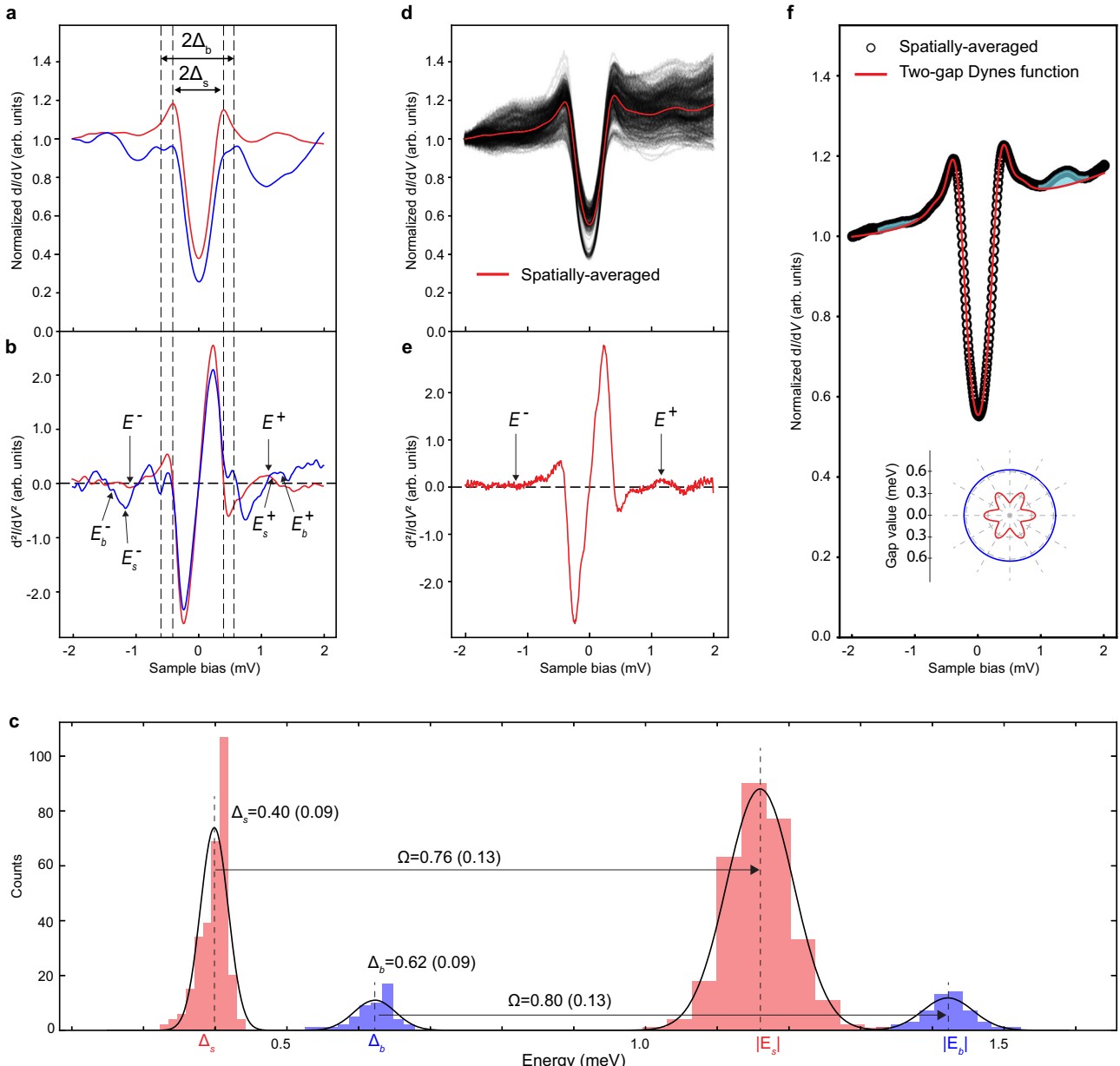

**Fig. 2 | Analysis of the SC gaps and bosonic mode energy in pristine CsV₃Sb₅.**
**a** Two types of representative d$I$/d$V$ spectra. The bigger and smaller SC gaps corresponding to the peak-to-peak distances are labeled by $\Delta_b$ and $\Delta_s$, respectively. **b** The corresponding derivative spectra d²$I$/d$V$² of (**a**), where the energies of the SC gaps and the bosonic mode offset by the two gaps can be quantitatively extracted. Labels in (**b**) are defined as $E_{s,b} = \pm(\Delta_{s,b} + \Omega)$, where $\Omega$ is the energy of bosonic mode. **c** Histograms of $\Delta_s$, $\Delta_b$, $|E_s|$, and $|E_b|$, plotted with the same counts of bins and fitted by normal distributions. The averaged values are $\Delta_s = 0.40(0.09)$ meV, $\Delta_b = 0.62(0.09)$ meV, $|E_s| = 1.16(0.09)$ meV, and $|E_b| = 1.42(0.09)$ meV, where the error is based on the energy resolution. The extracted bosonic mode energy is 0.76(0.13) meV from $|E_s|$ and $\Delta_s$, and 0.80(0.13) meV from $|E_b|$ and $\Delta_b$. **d** A series of

d$I$/d$V$ spectra obtained on the Cs and Sb surfaces of CsV₃Sb₅ in different regions obtained under the tunneling conditions $V_s = -2.0$ mV, $V_{mod} = 0.1$ mV, $I_t = 1$ nA. The spatially-averaged spectrum is highlighted by the red solid line. **e** The derivative of the spatially-averaged d$I$/d$V$ curve, from which the energies of the bosonic mode offset by the SC gaps ($E^+$ peak and $E^-$ dip in d²$I$/d$V$²) can be determined ($\bar{\Omega} = 0.76$ meV, $\bar{\Delta} = 0.40$ meV). **f** Two-gap Dynes functions description of the spatially-averaged d$I$/d$V$ spectrum isolated from (**d**), showing good overall agreement. The difference between the two-gap Dynes functions and the experimental data demarcates the contribution due to the bosonic mode, which is marked by the shadowed region with cyan color. The two gap functions are shown as polar plots in the inset of (**f**).

The corresponding Fourier transforms show weakening of the $2a_0 \times 2a_0$ CDW peaks (marked by the red circles) with increasing $x$, and the CDW order is fully suppressed at $x = 0.40$ (Fig. 3 and Supplementary Fig. 5). We collect a series of d$I$/d$V$ spectra over the SC gap energies from different regions on each CsV₃₋ₓTaₓSb₅, and show the stack plots in Fig. 3d–f. In contrast to undoped CsV₃Sb₅, the nonuniformity inside the SC gap is reduced, but remains significant at the energies of the coherence peaks and beyond. The peak-to-peak distance $\Delta$ and the peak height increase with $x$ as shown in Fig. 3d–f, indicating the enhancement

of the SC gap and the superfluid density[56]. Concurrently, the in-gap density of states is depleted progressively and a U-shape SC line-shape emerges. The corresponding spatially-averaged d$I$/d$V$ spectrum is superimposed in Fig. 3d–f and replotted in Fig. 3g–i. The peak-dip-hump features, i.e. the bosonic mode, are clearly visible at all different Ta-substitutions (Fig. 3g–i).

Remarkably, the spatially-averaged d$I$/d$V$ spectra in Fig. 3g–i continue to be well-described by the two-gap Dynes functions as in undoped CsV₃Sb₅, amid a systemic evolution of the gap functions from

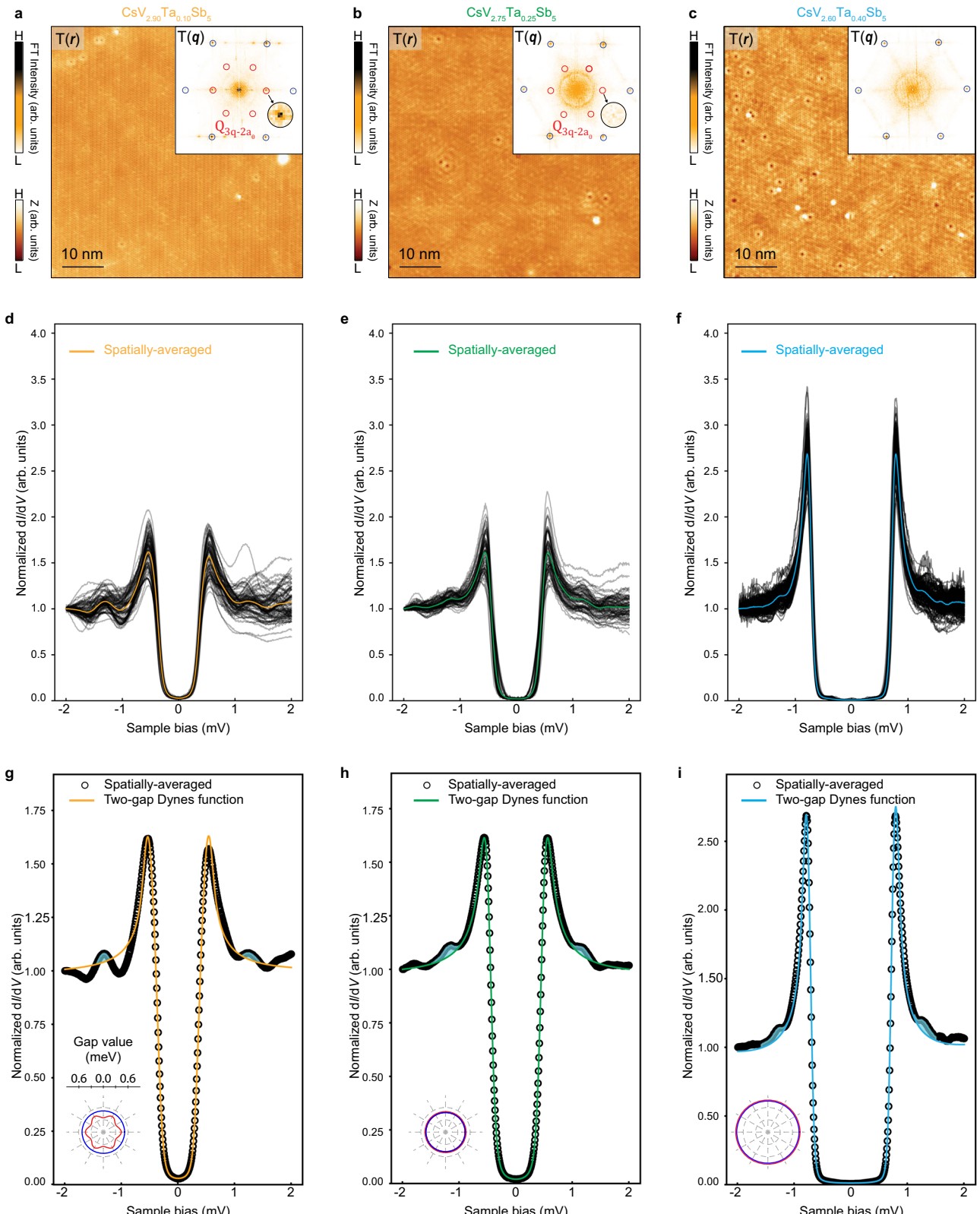

**Fig. 3 | Analysis of superconducting gaps and the bosonic mode in Ta-doped CsV₃Sb₅ (CsV₃₋ₓTaₓSb₅, x = 0.10, 0.25 and 0.40).** STM images obtained on the Sb surface of $CsV_{2.90}Ta_{0.10}Sb_5$ (**a**), $CsV_{2.75}Ta_{0.25}Sb_5$ (**b**), and $CsV_{2.60}Ta_{0.40}Sb_5$ (**c**), respectively ($V_s = -100$ mV, $I_t = 1$ nA). The corresponding Fourier transform (inset) shows the disappearing of $2a_0 \times 2a_0$ CDW with increasing of $x$. A series of d$I$/d$V$ spectra obtained in different region on the Cs and Sb surface of $CsV_{2.90}Ta_{0.10}Sb_5$ (**d**), $CsV_{2.75}Ta_{0.25}Sb_5$ (**e**), and $CsV_{2.60}Ta_{0.40}Sb_5$ (**f**) under the tunneling conditions $V_s = -2$ mV, $V_{mod} = 0.05$ mV, $I_t = 1$ nA, where the spatially-averaged spectra are highlighted in each stack plot by

orange, green and sky-blue colors, respectively. Two-gap Dynes functions description of the spatially-averaged d$I$/d$V$ spectra of $CsV_{2.90}Ta_{0.10}Sb_5$ (**g**), $CsV_{2.75}Ta_{0.25}Sb_5$ (**h**), and $CsV_{2.60}Ta_{0.40}Sb_5$ (**i**), respectively. The experimental data (black circles) matches very well with the two-gap Dynes functions curves (colored solid lines). The features of bosonic modes manifest in their difference just outside the SC gaps marked by the shadowed region with cyan color. The two gap functions are shown as polar plots in the insets of (**g**–**i**). The red circles and blue circles in (**a**–**c**) represent the wave vectors of $2a_0 \times 2a_0$ CDW and Bragg peaks, respectively.

having one anisotropic gap to having two isotropic gaps with nearly equal amplitudes. For $CsV_{2.90}Ta_{0.10}Sb_5$, the anisotropy of $\Delta_1$ is reduced, and $\Delta_1$ and isotropic gap $\Delta_2$ move closer to each other compared to the undoped $CsV_3Sb_5$ (Fig. 3g and Supplementary Table 1). At $x = 0.25$, the averaged d$I$/d$V$ spectrum is well-described by two isotropic gap functions of nearly the same size but different pair-breaking strengths ($\Delta_1 = 0.487$ meV and $\Gamma_1 = 0.071$ meV) and ($\Delta_2 = 0.478$ meV and $\Gamma_2 = 5 \times 10^{-4}$ meV) (Method and Supplementary Fig. 6). Increasing $x$ to 0.40, the two-gap Dynes functions show excellent agreement with the average d$I$/d$V$ line-shape for ($\Delta_1 = 0.782$ meV and $\Gamma_1 = 0.031$ meV) and ($\Delta_2 = 0.758$ meV and $\Gamma_2 = 2 \times 10^{-5}$ meV) in Fig. 3i. The two isotropic gaps have increased significantly but stay close to one another in magnitude. This agrees with the recent ARPES experiments on $CsV_{3-x}Ta_xSb_5$ at a similar $x$ value, including the size of the isotropic SC gaps[55,57]. Intriguingly, the pair-breaking strengths at much lower temperature (~50 mK) remain different even in the absence of the CDW order (Method and Supplementary Fig. 6), reflecting possibly the distinct orbital/band origin of the multi-gap functions.

Although the two-gap Dynes functions describe the low-energy density of states and the coherence peaks really well (colored solid lines in Fig. 3g–i), the spectral features beyond the coherence peak shoulders cannot be captured (shadowed regions in Fig. 3g–i). These excitations demonstrate the persistence of the collective bosonic modes in all Ta substituted $CsV_{3-x}Ta_xSb_5$ single crystals studied. It is important to note that the bosonic mode associated with the peak-dip-hump feature continues to be observable even when the long-range CDW order is fully suppressed at $x = 0.40$ (Fig. 3i). The Ta substitution causes the spectral weight of the coherence peak to increase significantly and lower the mode energy simultaneously. Together they make the signature of the bosonic mode weaker in appearance. As demonstrated for the undoped $CsV_3Sb_5$ above, taking the derivative of the spatially-averaged d$I$/d$V$ spectra (Fig. 3d–i), the mode energies can be determined approximately at the peaks (and dips) in the d$^2I$/d$V^2$ spectra for different Ta-substitution ratio $x$ (Supplementary Fig. 7).

## Discussion

Using ultra-high energy resolution STM/S at ultra-low temperature, we discovered a collective bosonic mode coexisting with two-gap superconductivity in kagome superconductors $CsV_{3-x}Ta_xSb_5$. It is likely that the isotropic gap with a smaller pair-breaking effect comes from the circular Fermi surface of Sb-$p_z$ band around the center of Brillouin zone, and the anisotropic gap with a bigger pairing-breaking effect originates from the V-$d$ bands near the zone boundary[41,43,54]. The evolutions of $\Delta_{1,min}$, $\Delta_{1,max}$, and $\Delta_2$ are plotted as a function of increasing Ta-substitution $x$ in Fig. 4a. The two SC gaps, with different pair-breaking strengths and different orbital/band characters, move closer in energy, merge into two isotropic gaps of equal amplitude, and then increase synchronously at large $x$ with the suppression of CDW order. In Fig. 4b, we plot the ratio $2\Delta_{1,max}/k_BT_c$ and $2\Delta_2/k_BT_c$ versus $x$, where the SC transition temperature $T_c$ is taken from the transport measurements. In the undoped $CsV_3Sb_5$, the large gap $\Delta_2$ is in the strong-coupling regime, but with a smaller pair-breaking strength $\Gamma_2$, while the anisotropic gap $\Delta_1$ with a larger pair-breaking strength $\Gamma_1$ supports a $2\Delta_{1,max}/k_BT_c$ close to the BCS weak-coupling value of 3.53[58]. With increasing $x$, $2\Delta_2/k_BT_c$ reduces while $2\Delta_{1,max}/k_BT_c$ holds steady such that they converge to the BCS value at large $x$.

We next turn to discuss the possible origin of the bosonic mode. In Fig. 4c, the mode energy $\Omega$ is plotted as a function of the Ta-substitution $x$. It decreases continuously with increasing $x$ accompanied by the increase of the SC gap $\Delta$ defined by the peak-to-peak distance in the conductance spectra and the suppression of the CDW order (Fig. 3). Such an anti-correlation between the Leggett mode and the superconducting gap has also been observed in monolayer NbSe$_2$[14]. The Leggett mode is expected to be proportional to the SC

gaps but anti-correlated with the density of states[12]. When the CDW order in kagome superconductors is suppressed, the normal state density is likely to receive a more significant enhancement than the superconducting gap, resulting in an anti-correlation relationship. To compare the mode energy with the pair-breaking or quasi-particle excitation energy $2\Delta$, we plot $\Omega/2\Delta$ as a function of $x$ in Fig. 4d. The mode energy is close to $2\Delta$ in the parent $CsV_3Sb_5$ ($x = 0$) and moves to far below $2\Delta$ with Ta-substitution $x$ (Fig. 4d).

The fact that this bosonic mode exists in the absence of CDW order at $x = 0.40$ rules out the possibility that it is associated with a collective mode of the CDW order[9]. The spectral weight of the Raman mode observed in $CsV_3Sb_5$ peaks at 5.5 meV in energy[59,60], which is well above the bosonic mode energy. We approximately rule out phonons as an origin since there has been no observed phonon modes at much lower energies than Raman mode in previous STM works on superconductors. Unlike in the high-$T_c$ cuprates where strong spin fluctuations can give rise to a collective mode of magnetic origin[61], these kagome metals are nonmagnetic and the spin fluctuations are much weaker than charge fluctuations, as indicated by nuclear magnetic resonance measurement[62] in $CsV_3Sb_5$. In principle, a spin exciton resonance mode can emerge in the SC state with the opening of the SC gap, similar to a semiconductor, but the weak spin-spin correlation cannot provide enough binding energy to pull the mode energy well below the pairing-breaking energy. Thus, the bosonic mode is most likely a low-energy collective mode associated with the SC order parameter fluctuations in these unique nonmagnetic multi-band kagome superconductors.

The SC modes involve the phase and amplitude fluctuations[6–12]. The amplitude mode, i.e. the Higgs mode[7,9] (depicted in Fig. 4e) usually resides close to the pair-breaking energy $2\Delta$. The small values of $\Omega/2\Delta$ at large $x$ therefore suggest the mode cannot be a pure Higgs mode in $CsV_{3-x}Ta_xSb_5$. The massless global phase mode (Nambu–Goldstone mode) is pushed to the plasma energy well above $2\Delta$ by Coulomb interaction through the Anderson-Higgs mechanism[7]. However, in multi-band superconductors, the relative phase mode or the Leggett mode[12] is a low-energy mode that is associated with the transfer of Cooper pairs from one band to another. For a two-component superconductor described by two order parameters, $\Delta_1 e^{i\phi_1}$ and $\Delta_2 e^{i\phi_2}$, the Leggett mode associated with the phase difference $\phi_1 - \phi_2$ is depicted in Fig. 4e by the green and purple arrows. The two-components are generally coupled by the Josephson coupling $J$, which generates a mass for the Leggett mode $\propto \sqrt{J\Delta_1\Delta_2}$[12]. The quasi-particle density of states is renormalized by the coupling to the collective mode, which results in the dip-peak-hump feature in d$I$/d$V$ spectra measured by STM/S, as has been explicitly shown by calculations for the Leggett mode in the context of NbSe$_2$[14].

Since at large $x$, the bosonic mode, with energy $\Omega$ well below the pair-breaking energy (Fig. 4d), is thus naturally identified with a dominant Leggett mode. In general, the Leggett mode mixes with the amplitude Higgs mode by the Josephson coupling and other effects such as pair-breaking scattering, nonuniformity, and thermal excitations in real materials. Therefore, the collective bosonic mode observed here can be referred to as a Higgs-Leggett mode. As $x$ is reduced, the tunneling conductance spectrum deviates more significantly from that of an ideal BCS superconductor and the value of $\Omega/2\Delta$ (Fig. 4d) increases possibly due to an enhanced Josephson coupling between different SC components. In addition, with the onset of CDW order at smaller $x$, the 3Q PDW order emerges[40] with different components coupled by Josephson coupling[45]. It is thus reasonable to expect a stronger mixing between the Higgs and Leggett mode. At $x = 0$, the bosonic mode energy is just below the single-particle excitation gap (Fig. 4d), consistent with a dominant Higgs mode pulled down from $2\Delta$ by the coupling to the Leggett mode. It will be interesting to study the evolution of this distinct bosonic mode using optical measurement[16,63].

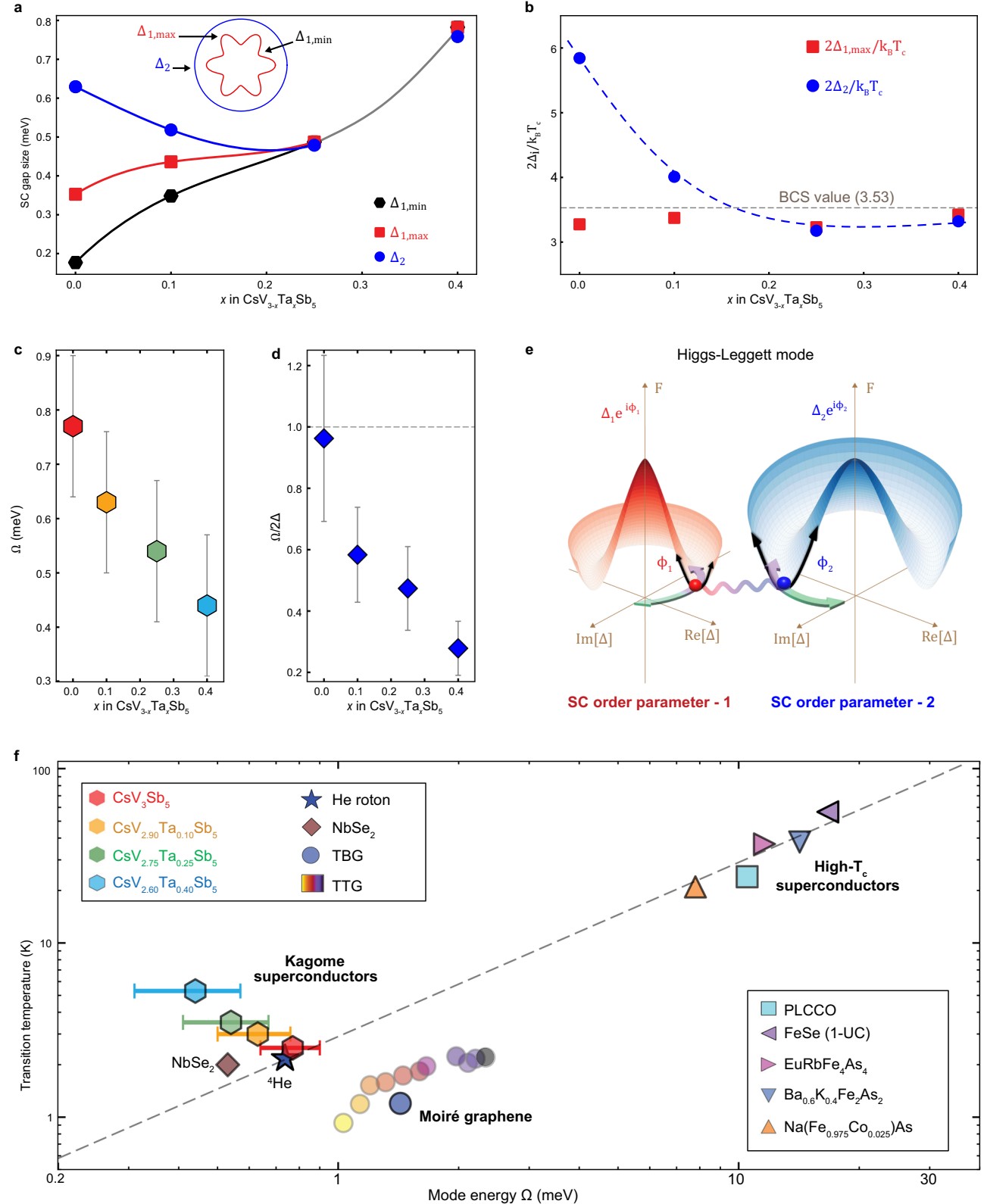

A more exotic possibility is that the bosonic mode is the Bardasis-Schrieffer mode[64], which can arise due to the fluctuations of a subleading pairing order parameter, e.g. competing $d$-wave pairing fluctuations in a dominant $s$-wave superconductor. The energy of such an exciton-like mode in the particle-particle channel lies below the pair-breaking energy of the dominant pairing and reduces as the pairing strengths in the subleading and the leading pairing angular momentum channels approach one another. Signatures of the Bardasis-

Schrieffer mode have indeed appeared in multi-band Fe-based superconductors[65–67], where a subleading pairing of $d$-wave symmetry in a dominant $s$-wave SC state is likely present. For the kagome superconductors studied here, in addition to the above, an intriguing possibility is that the two gap functions $\Delta_1$ and $\Delta_2$ originate from two order parameters of different symmetry, e.g. an anisotropic $d + id$ pairing of the V band near the zone boundary as proposed theoretically[45,68] and an isotropic $s$-wave pairing of the Sb $p_z$ band near

**Fig. 4 | The evolution of SC gaps and the collective mode energy in CsV$_{3-x}$Ta$_x$Sb$_5$.**
**a** Plot of $\Delta_{1,min}$, $\Delta_{1,max}$ and $\Delta_2$ (marked in the inset) determined from the spatially-averaged spectrum as a function of the Ta-substitution ratio $x$. $\Delta_{1,min}$ and $\Delta_{1,max}$ are the minima and maxima SC amplitude of the SC gap 1, and $\Delta_2$ is the SC amplitude of the SC gap 2. **b** The gap to $T_c$ ratio, $2\Delta_2/k_BT_c$ and $2\Delta_{1,max}/k_BT_c$, as a function of Ta-substitution $x$, showing that $2\Delta_2/k_BT_c$ is large and indicative of a strong-coupling superconductor in the undoped case[40], decreases and approaches to the BCS value (3.53) with increasing $x$, while $2\Delta_{1,max}/k_BT_c$ is always close to BCS value for all $x$. $k_B$ is the Boltzmann constant. **c** Scatter plot of the bosonic mode energy $\Omega$ versus Ta-substitution ratio $x$, showing that $\Omega$ decreases with increasing $x$. **d** Plot of $\Omega/2\Delta$ versus Ta-substitution ratio $x$, showing $\Omega$ is far below the pair-breaking energy $2\Delta$ at large $x$, where $\Delta$ is the SC gap measured by the coherence peak-to-peak distance. In

pristine CsV$_3$Sb$_5$, $\Delta$ is effectively determined by the smaller SC gap. **e** Schematics of the Higgs-Leggett mode, showing the amplitude and phase fluctuation of the SC order parameters. The Higgs mode is marked by black arrows. The Leggett mode ($\phi_1$-$\phi_2$) is marked by the wavy line and colored arrows, where $\phi_1$ and $\phi_2$ are the SC phase. **f** Plot of transition temperature $T_c$ as a function of mode energy $\Omega$. The data are taken or extracted from the STM works on kagome superconductors reported here, NbSe$_2$[14], twisted bilayer graphene[70], twisted trilayer graphene[71], Pr$_{0.88}$LaCe$_{0.12}$CuO$_4$[3], one-unit-cell FeSe[4], EuRbFe$_4$As$_4$[72], Ba$_{0.6}$K$_{0.4}$Fe$_2$As$_5$[48,49], and Na(Fe$_{0.975}$Co$_{0.025}$)As[48], as well as from the Uemura plot[5]. The $T_c$s of twisted trilayer graphene are extracted from ref. 80. The dashed line in (**f**) indicates the Uemura line ($k_BT_c \sim \Omega/4$). The error bars in (**c**, **d**) are based on the energy resolution statistical deviation.

the zone center. In this case, the Josephson coupling between the two SC components is suppressed, but each is serving as the subleading pairing channel of the other in the $s + d + id$ state. The fluctuations of the order parameters are expected to produce the Bardasis-Schrieffer mode as have been shown for the $s + id$ state theoretically[69], which is not inconsistent with our observations. While we cannot pin down between the Higgs-Leggett mode or the Bardasis-Schrieffer mode by STM/S, our discovery of the SC mode and its evolution with the suppression of the CDW order points to probing the spatial symmetry of the collective mode by low-energy Raman spectroscopy which can potentially uncover the pairing symmetry of the kagome superconductors. Further work is still necessary to confirm the origin of this distinct bosonic mode.

In summary, we have observed a distinct collective mode of SC fluctuations in kagome superconductors CsV$_{3-x}$Ta$_x$Sb$_5$ in a SC state well-described by an isotropic gap and an anisotropic gap when $x$ is small. The mode energy remains below the threshold for single-particle excitations and reduces concomitantly with the suppression of CDW order and enhancement of the SC gap (peak-to-peak distance) with increasing Ta-substitution. In Fig. 4f, we plot the SC transition temperature as a function of the bosonic mode energy $\Omega$ determined or extracted from STM/S measurements. In addition to the kagome superconductors study here, we also include available data from single-layer NbSe$_2$[14], Morie graphene[70,71] and some iron- and copper-based high-$T_c$ superconductors[3,4,48,49,72] in such a Uemera-like plot. The neutron data for the roton mode in superfluid $^4$He[73] are also included. The data points in Fig. 4f fall into two groups, high-$T_c$ superconductors and low-$T_c$ superconductors. For most of high-$T_c$ superconductors, the STM bosonic mode follows the Uemura line[5] ($k_BT_c \sim \Omega/4$). While the stoichiometric CsV$_3$Sb$_5$ still lies close to this line, as does NbSe$_2$ film, Ta substitution systematically drives the ratio $\Omega/k_BT_c$ away from the Uemura value. This is likely due to the nature of the low-lying SC collective excitations. Our findings provide insights for understanding kagome superconductivity and its interplay with other correlated states of matter.

## Methods
### Single crystal growth of CsV$_{3-x}$Ta$_x$Sb$_5$
Single crystals of CsV$_3$Sb$_5$ were grown via the modified self-flux method and characterized as presented in ref. 40. The electronic temperature was determined as described in more detail in ref. 40. Single crystals of Ta-substituted CsV$_{3-x}$Ta$_x$Sb$_5$ were synthesized from Cs liquid (Alfa, purity 99.98%), V powder (Alfa, purity 99.9%), Ta powder (Alfa, purity 99.98%) and Sb shot (Alfa, purity 99.999%) via a modified self-flux method. The mixture was placed into an alumina crucible and then sealed in a quartz ampoule under a high vacuum atmosphere. Subsequently, the sealed quartz ampoule was heated to 1100°C, held for 72 hours, and gradually cooled down to 500°C at a rate of 2°C per hour. Finally, the single crystals were separated from the flux. Due to the high reactivity of the Cesium, all preparation procedures were carried out in an argon-filled glovebox, except for the sealing and reaction procedures.

### STM/ STS
The samples used in the experiments were cleaved in situ at 15 K and immediately transferred to an STM chamber. Experiments were performed in an ultrahigh vacuum ($1 \times 10^{-10}$ mbar) ultra-low temperature STM system ($T_{base}$ ~ 50 mK, $T_{electron}$ ~ 300 mK) equipped with 9-2-2 T magnetic field. All the scanning parameters (setpoint voltage and current) of the STM topographic images are listed in the captions of the figures. Unless otherwise noted, the differential conductance (d$I$/d$V$) spectra were acquired by a standard lock-in amplifier at a modulation frequency of 973.1 Hz, the second-order differential conductance (d$^2I$/d$V^2$) spectra and the negative third-order differential conductance (-d$^3I$/d$V^3$) spectra were the numerical derivative of the d$I$/d$V$ spectra. Tungsten STM tip was fabricated via electrochemical etching and annealed to bright orange color. The electron temperature $T_{electron}$ in the low-temperature STS is calibrated using a standard superconductor, Nb crystal.

### Fabrication of the superconducting CsV$_3$Sb$_5$ nanoflake tip
The tungsten STM tip was used to inject the clean surface of the sample more than 15 nm depth, holding for 5–10 s with the voltage 1–2 V, and withdraw to its original position. After that, the CsV$_3$Sb$_5$ nanoflake will stick to the apex of the tungsten tip. We usually obtained the stable CsV$_3$Sb$_5$ nanoflake tip after repeating the "injection process" for many times at several clean as-cleaved surface regions.

### Discussion on the Leggett mode in Josephson tunneling spectroscopy
CsV$_3$Sb$_5$ is a two-band superconductor, so two types of Josephson tunneling, namely "intra-band" and "inter-band" Josephson tunneling (Supplementary Fig. 4a), will contribute to the experimental signal when the CsV$_3$Sb$_5$ nanoflake tip is used. This scenario is in analogy to the well-known "Fujita conjecture" proposed for cuprates[74]. As shown in Supplementary Fig. 4a, the "intra-band" Josephson tunneling is between the same band in the sample and the tip, and the "inter-band" Josephson tunneling is between different bands in the sample and the tip. In this scenario, the Leggett mode is expected to enhance the tunneling current when the energy is greater than $\Omega$, which is manifested as a peak in the d$I$/d$V$ spectrum. Supplementary Fig. 4b shows the signature of the Leggett mode at $|E| = \Omega$.

### Extraction of peak-to-peak superconducting gap size and mode energy in CsV$_3$Sb$_5$
In most cases, the two SC gaps are not easily identifiable precisely and unambiguously by STM/S, although macroscopic measurements[42] support a two-gap structure of superconductivity in CsV$_3$Sb$_5$. Frequently, a dominated smaller SC gap accompanied by a weak bigger SC gap is observed concomitant with one pair of peak-dip-hump feature at energy $|E^{\pm}|$ (Figs. 1d and 2e). In total, there are three types of d$I$/d$V$ spectra observed in our experiments, including one-gap dominated d$I$/d$V$ spectra, resolved two-gap d$I$/d$V$ spectra, and smeared two-gap d$I$/d$V$ spectra. (1) For the one-gap dominated curves, we can clearly identify the (smaller) SC gap and its

corresponding bosonic mode. (2) For the two-gap distinguishable d$I$/d$V$ spectra, we can also clearly identify the two SC gaps and their corresponding bosonic modes. (3) For the smeared two-gap d$I$/d$V$ spectra, where the two-gap structures are visualized but without significant resolution, we apply a negative second derivative technique (-d$^3I$/d$V^3$) on the d$I$/d$V$ spectra. It can magnify the obscured peak (or dip) signals in the original curve (Supplementary Fig. 1a), as utilized to identify the pseudogap in the cuprate superconductors[75] and Andreev reflection peaks in amorphous indium oxide[76]. In that case, two broad coherence peaks in d$I$/d$V$ spectra are resolved into two pairs of sharp peaks in the -d$^3I$/d$V^3$ spectrum, which indicates the detection of two superconducting gaps $\Delta_s$ and $\Delta_b$ (Supplementary Fig. 1b). Statistically, we observe the one-gap dominated curves with the highest probability.

## Dynes functions analysis

We describe the tunneling density of states using the Dynes function $D_1(E)$ for an anisotropic gap function with $C_6$ symmetry $\Delta_{1,max}\cos^2 3\theta + \Delta_{1,min}\sin^2 3\theta$,

$$D_1(E) = \int_0^{2\pi} d\theta \, \mathrm{Re}\left[\frac{E - i\Gamma_1}{\sqrt{(E - i\Gamma_1)^2 - \left[\Delta_{1,max}Cos^2 3\theta + \Delta_{1,min}Sin^2 3\theta\right]^2}}\right]$$

(M1)

where the pair-breaking parameter $\Gamma_1$ is assumed to be isotropic. This works reasonably well at describing the d$I$/d$V$ spectrum dominated by a single-pairs of coherence peaks and a V-shaped gap[53]. Since two SC gaps are visualized in CsV$_3$Sb$_5$ (Fig. 2a), we include an additional contribution $D_2(E)$ associated with an isotropic gap $\Delta_2$ and pair-breaking $\Gamma_2$,

$$D_2(E) = Re\left[\frac{E - i\Gamma_2}{\sqrt{(E - i\Gamma_2)^2 - \Delta_2^2}}\right]$$

(M2)

The total tunneling density of states is given by,

$$D(E) = (1 - \alpha) \times D_1(E) + \alpha \times D_2(E)$$

(M3)

where $\alpha$ controls the relative spectral weight between $D_1(E)$ and $D_2(E)$. The two-gap Dynes functions description can therefore capture the contributions from a second gap of a much weaker spectral intensity, possibly from a different orbital, in the elevated tails outside the coherence peaks of the spectrally-dominating single-gap.

The final formula for the tunneling conductance d$I$/d$V$ spectrum is quite standard[77–79],

$$S(E) = -[P(E) \cdot (D(E) + A)] \otimes f'(E, T) \otimes b(E) \otimes G(E)$$

(M4)

where $f'(E, T)$ is the derivative of Dirac-Fermi function at temperature T, $\otimes$ is the energy convolution operator, and the functions $P(E), b(E)$, and $G(E)$ account for the normal state density of states, the machine resolution, and the average over spatial distributions, respectively. More specifically, we use a parabolic function $P(E) = a + bE + cE^2$ to approximate the normal state density of states at low-energies. A constant $A$ was also introduced to represent the gapless conductance, which could stem from impurity-induced localized states or excitations of the PDW[40]. Assuming the spatially distribution of d$I$/d$V$ spectrum is uncorrelated, a Gaussian function $G(E) = \exp(-E^2/2\sigma^2)$ is used to represent the quenched average over disorder, where the width $\sigma$ measures the disorder strength. In addition to the thermal broadening described by the Fermi-Dirac function at an electron temperature $T_{electron} = 0.3$ K[40], we also take into account our instrument resolution due to the lock-in technique, which

is describe by $b(E)$[78],

$$b(E) = \begin{cases} \frac{2}{\pi V_{mod}}\sqrt{1 - \left(\frac{E}{V_{mod}}\right)^2}, & (|E| < V_{mod}) \\ 0, & (|E| > V_{mod}) \end{cases}$$

(M5)

where $V_{mod}$ is the lock-in modulation.

## Determination of parameters in the Dynes functions analysis

In practice, we adapt the following procedures in the two-gap Dynes functions analysis:

1. A single anisotropic Dynes function $D_1(E)$ is used to describe the experimental data, where the SC gap bottom and coherence peaks are controlled by $\Delta_1$, the anisotropy parameter $\tau$, and the pair-breaking strength $\Gamma_1$.
2. The second isotropic Dynes function $D_2(E)$ is then included to reduce the discrepancies between the experimental data and the single-gap Dynes function $D_1(E)$ within the energy of SC coherence peak shoulders where $\alpha$ and $\Gamma_2$ are adapted. In this step, $\Gamma_2$ is gradually increased to get the best description of experimental data, where the comparisons of different choices of $\Gamma$s are shown in Supplementary Fig. 6.
3. A parabolic function $P(E)$ is used to match the normal state density of states outside the superconducting gaps.
4. A Gaussian function $G(E)$ is used to optimize the difference compared to experimental data.

All parameters for the two-gap Dynes functions analysis are listed in Supplementary Table 1.

## Data availability

Relevant data supporting the key findings of this study are available within the article and the Supplementary Information file. All raw data generated during the current study are available from the corresponding authors upon request.

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

## Acknowledgements
The work is supported by grants from the National Natural Science Foundation of China (62488201), the National Key Research and Development Projects of China (2022YFA1204100 and 2019YFA0308500), the Chinese Academy of Sciences (YSBR-003) and the Innovation Program of Quantum Science and Technology (2021ZD0302700). Z.W. is supported by the US DOE, Basic Energy Sciences Grant No. DE-FG02-99ER45747 and the Cottrell SEED Award No. 27856 from Research Corporation for Science Advancement.

## Author contributions
H.-J. G. and Z.W. supervised the project. H.-J.G. designed the experiments. B.H., H.C., Y.Y. and H.X. performed STM experiments and analyzed data with assistance of Z.H., X.H. and X.L. H.Y. Z.Z prepared samples. Z.W. did the theoretical consideration. All of the authors participated in analyzing the experimental data, plotting figures and writing the manuscript.

## Competing interests
The authors declare no competing interests.
