## [Peer Review File · Nature Communications]

REVIEWER COMMENTS

Reviewer #1 (Remarks to the Author):

In the revised version of the manuscript, the authors have addressed the issues that I pointed out in the previous report. Especially, they included additional data (Extended Data Fig. 3) showing the temperature dependence of dI/dV spectra with both normal and superconducting STM tips. It is clearly seen that the peak-dip-hump structure disappears for $T > T_c$, indicating that it is correlated with the superconducting order. These data are very encouraging and interesting. How does the bosonic mode energy change with temperature? Does it approach zero as T gets closer to T_c (as expected in the case of Leggett mode)? The anti-correlation between the mode energy and doping level is also addressed from the point of view of density of states. This could be a possible explanation. In the reply to the referee report, it is also clearly explained how to determine E^{\pm} from the dI/dV curve. In the revised manuscript, it says that “Those dips/peaks are chosen by prominent, closest to the superconducting gaps, and particle-hole symmetry”, but this sentence sounds unclear. Can the authors improve this sentence? The sentence in the report is much better: “we choose the most prominent and symmetric local minimal/maximal that is closest to the coherence peak, while the other peaks/dips may be the data noise due to the weakness and particle-hole asymmetry”.

To summarize, the authors' response to my previous comments is quite satisfactory. The paper is significantly improved, and positive support for the superconducting collective mode is obtained from the additional data. Based on these, I would like to recommend the publication of the paper in Nature Communications.

Reviewer #2 (Remarks to the Author):

Since the previous version, the authors have significantly improved their paper by adding new temperature dependence data, tunnelling spectra with superconducting tip and histograms for gap energy and mode energy in larger FOV. They also properly weaken the claim about γ in the Dynes fittings. I think the new data make the conclusion more solid, which adds great value to this paper. Now, the paper provides at least a self-consistent observation of Leggett mode in this kagome superconductor family. I can support its publication in Nature Communications.

Reviewer #3 (Remarks to the Author):

<see attachment for color code>

I appreciate the authors for responses to my comments. I am also grateful to the authors for additional Josephson Tunneling measurements in the responses. While Fig. R1c captures the signature of the Josephson Tunneling, this looks just like a conventional Josephson Tunneling, and I don't see a clear signature associated with the Leggett mode in the data. Please see details below. I have to say that the manuscript lacks experimental evidence for the presence of the Leggett mode. I don't think that the peak-dip-hump structure in the local density of states can be evidence of the Leggett mode in the present study. Thus, I can't recommend for a publication in the Nature Communications. My responses are in green.

Comment 3: The authors just analyzed ~300 dI/dV spectra for each x. First, it's not clear where these spectra are taken exactly in the field of view. Second and more importantly, it's not trivial that ~300 spectra are statistically enough to account for variations in the spectral shape (or heterogeneity in the electronic structure). It's possible that these spectra may not be representative but may be selected from local regions that is not an intrinsic property in this compound. Obviously, there are variations in the spectral shape as shown in Fig. 2 and 3, and the authors should demonstrate that ~300 spectra are enough. For example, one can take a dI/dV map in 50nm FOV with 256×256 pixels at given energy and construct a histogram of the values in dI/dV map. Then, one can simply crop/divide this map into four regions (4×25 nm² 128×128 pixels maps) and construct the histogram again for each map. If the size of the FOV and pixels# are statistically enough, then I would expect that widths of the distribution are the same before and after the crop. Alternatively, and more convincingly, the authors can take a data in the same way as in the Ref. 2, constructing both " and # maps and visualizing spatial structure of them.

Response 3: We thank the reviewer for the suggestion on constructing $\Delta(r)$ and $\Omega(r)$ maps. New Extended Data Fig.2 (Fig. R3) in the revised manuscript shows the result of $\Delta(r)$ and $\Omega(r)$ collected within a field of view of 25× 25 nm and 180×180 pixels. The spatial variation and fertile patterns are visualized clearly, implying the spatial variation of dI/dV spectra in CsV3Sb5.

Fig. R3. Spatial distribution of $\Omega(r)$ and $\Delta(r)$ of CsV3Sb5. a, Spatial distribution of $\Delta(r)$ obtained on the Sb surface within the field of view of 25 nm × 25 nm, showing the spatial inhomogeneity of superconducting gap ($V_s = -2.0$ mV, $I_t = 1$ nA, $V_{mod} = 0.1$ mV). b, The histogram plot of a. c, Spatial distribution of $\Omega(r)$ obtained on the same field of view of a, showing the spatial inhomogeneity of the bosonic mode. d, The histogram plot of c.

Accordingly, we added spatial distribution of $\Delta(r)$ and $\Omega(r)$ as new Extended Data Fig. 2 and revised the statement to clarify the methodology in the revised manuscript, as follows:

“In Fig. 2c, we show a series of dI/dV curves with the dominant one-gap line-shape in a stackplot without offset from different regions and samples. The spatial distribution of $\Delta(\&)$ and $\Omega(\&)$ within a field of view of 25 nm × 25 nm are presented in Extended Data Fig. 2.”

I understand the statistics. But it looks to me that there is a tip change in the data, and data quality at the top 25% of the FOV is quite different where the gap values are highly uniform around 0.35meV. This would largely contribute to the histogram in Fig. R3b. Nevertheless, a full width the half maximum is about 0.02meV that is about 6% from its average. This suggests that the Δ is highly uniform compared to other compounds such as those in cuprates.

Comment 6: A Leggett mode energy is proportional to the superconducting order parameter amplitude (S.G. Sharapov et al., *Eur. Phys. J. B* 30, 45–51 (2002)). Thus, I would expect that the mode energy Ω should increase with increasing the gap size Δ , unless a coupling constant changes dramatically with x . However, as shown in Fig. 4, Ω and Δ show an opposite trend - Ω decreases but Δ increases with x . Why do they show the opposite trend with x ?

Response 6: We thank the reviewer for this critical comment. In the primary work studied by Professor A. J. Leggett (*Prog. Theor. Phys.* 36, 901-930 (1966)), the mode energy expressed as:

where $\rho_{1,2}$ is the density of states at Fermi surface, U and V are the pairing potential, J is the Josephson coupling between the two superconducting components, $\Delta_{1,2}$ is the superconducting order parameter. It is natural to expect a positive correlation between the superconducting gap (or T_c) and mode energy when assuming an unchanged density of states. However, with the introduction of Ta substitution, the normal state density will get an enhancement, stemming from the weakening of the charge density wave. When the change in the normal state density is much significant than in the superconducting gap, an anti-correlation between the superconducting gap and bosonic mode energy emerges. It should be also noted that the anti-correlation between superconducting gap and Leggett mode energy has been well-studied in single-layer NbSe₂ (*Adv. Mater.* 34, e2206078 (2022)).

To avoid confusion on the anti-correlation between the superconducting gap and Leggett mode energy emerges in the Kagome superconductors, we have added discussions in the revised manuscript as follows:

“We next turn to discuss the possible origin of the bosonic mode. In Fig. 4c, the mode energy Ω is plotted as a function of the Ta-substitution x . It decreases continuously with increasing x accompanied by the increase of the SC gap Δ defined by the peak-to-peak distance in the conductance spectra and the suppression of the CDW order (Fig. 3). Such an anti-correlation between the Leggett mode and the superconducting gap has also been observed in monolayer NbSe₂. The Leggett mode is expected to be proportional to the SC gaps but anti-correlated with the density of states¹². When the CDW order in kagome superconductors is suppressed, the normal state density is likely to gain a more significant enhancement than the superconducting gap, resulting in an anti-correlation relationship.

I am not convinced by these arguments. In general, in the CDW ordered phase, the Fermi surface is expected to be reconstructed and be (partially) gapped out. ρ_1 and ρ_2 are the density of states at the Fermi level, so they are supposed to be (partially) gapped out in the CDW ordered phase. So, I don't understand why ρ_1 and ρ_2 are enhanced by weakening the CDW states. ρ_1 and ρ_2 are expected to be independent on the magnitude of the CDW order parameter.

Comment 8: In line 276, 'The low-energy Raman modes observed in CsV3Sb5 is at least 5.5 meV in energy^{60,61}, which is well above the bosonic mode energy. We can safely rule out phonons as an origin since there has been no observed phonon modes at such low energies.'. In fact, Ref. 61 reported dispersions of the Longitudinal and Transverse acoustic phonons. I don't think that the author can rule out a possibility that one of these phonons are coupled to the electrons, such that the Bogoliubov quasiparticle dispersion is renormalized by the electron-phonon coupling, resulting in a peak-dip-hump structure in dI/dV spectra.

Response 8: We thank the reviewer for the thoughtful consideration. Despite the dispersive nature of longitudinal and transverse acoustic phonons, the spectral weight is notably concentrated at ~ 5.5 meV (Phys. Rev. X 11, 031050 (2021)). Consequently, we do not anticipate these phonons to effectively couple with electrons. In addition, the bosonic mode observed in our manuscript can exist when the CDW is totally suppressed by Ta substitution, thus we exclude its phonon origin. We also stress the argument in the manuscript as follows:

"The spectral weight of the Raman mode observed in CsV3Sb5 peaks at 5.5 meV in energy^{65,66}, which is well above the bosonic mode energy. We can safely rule out phonons as an origin since there has been no observed phonon modes at such low energies."

Raman can't measure a dispersion of the phonons. The energy measured by Raman would be a value near the zone edge in momentum transfer space or at a momentum transfer constrained by the symmetry in the measurement. A phonon that may couple to electrons can be at finite momentum transfer at different energy from 5.5 meV, depending on the band structure and the phonon dispersion. I don't think that the authors can rule out a possibility of the phonon origin.

Comment 9: Regarding the Fig. 4f, Ref. 2 reported that the peak-dip-hump structure is due to the electron-phonon coupling. However, there is virtually no change in T_c when 16O is substituted by 18O, while the lattice vibration frequency is shifted by the amount of a difference in the oxygen mass in Bi2212. It's also the fact that the Ω is doping independent from slightly underdoped to overdoped regime, while Δ decreases with increasing doping as reported in the Ref. 2. Bi2212 obviously deviates from the line in Fig. 4f.

Response 9: Yes, we have taken the data from Nature 442, 546-550 (2006) into Fig. 4f without introducing any bias. We have refrained from commenting on the behaviour of Ω in cuprate in our manuscript.

Nature 442, 546-550 (2006) eventually studied the doping dependence in addition to the isotope effect. It's strange to selectively pick up values only for the O18 and O16 samples in Fig. 4f. The authors should include the Ω values for different doping levels from this reference as the authors mention 'without introducing any bias'.

Comment 11: I don't understand why the authors divided the data points in Fig. 4f into two groups as in the figure. I can understand if these data points are grouped by those that follow the linear relation and those that deviate from the line. In fact, Bi2212 also deviates from this line, so I don't think the Bi2212 can be the same group as other high T_c superconductors in this plot. The manuscript is not much benefitted from the Fig. 4f.

Response 11: We express our gratitude to the reviewer for their thoughtful consideration. We believe that Fig. 4f in our manuscript provides the most informative representation and is plotted without bias. Notably, we abstained from attempting to cluster or group the data points, and the inclusion of the Uemura line is merely intended as a guiding reference. It is crucial to emphasize that we refrained from drawing additional conclusions based on Fig. 4f, despite the divergence of kagome superconductors, moiré materials, and cuprates from the dashed Uemura line. In the revised manuscript, we amended the statement regarding the high- T_c superconductors as follows:

“For most of high- T_c superconductors, the STM bosonic mode follows the Uemura line⁵ ($k_b T_c \sim \Omega/4$). While the stoichiometric CsV₃Sb₅ still lies close to this line, as does NbSe₂ film, Ta substitution systematically drives the ratio $\Omega/k_b T_c$ away from the Uemura value. This is likely due to the nature of the low-lying SC collective excitations.”

Obviously, as I pointed out, T_c - Ω relationship in Bi2212 strongly deviates from the dashed line in Fig. 4, although the authors didn't include the T_c - Ω values for different doping levels from the ref. 2 (J. Lee et al., Nature 442, 546-550 (2006)). Cuprates are effectively a single-band system, and no Leggett mode has been realized. Thus, the deviation from the dashed line doesn't necessarily imply an existence of the SC collective excitations.

Comment 13: I would suggest the authors to perform Josephson Tunneling Microscopy measurement using the superconducting tip. I would expect a feature associated with the Leggett mode in the Josephson current spectra if exists.

Response 13: We thank the reviewer for the insightful suggestion. We have constructed the Josephson tunnelling spectroscopy and investigated the bosonic mode further. Fig. R1 (New Extended Data Fig. 3) shows that the signal of bosonic mode becomes more significant by using the Josephson tunnelling spectroscopy, and the bosonic mode accompanied by the superconducting gap and Josephson current disappears when $T > T_c$, demonstrating a superconducting collective mode.

In the revised manuscript, we add the description on fabrication of superconducting STM tip in the method part as follows:

“Fabrication of the superconducting CsV3Sb5 nanoflake tip. The tungsten STM tip was used to inject the clean surface of the sample more than 15 nm depth, holding for 5~10 s with the voltage 1~2 V, and withdraw to its original position. After that, the CsV3Sb5 nanoflake will stick to the apex of the tungsten tip. We usually obtained the stable CsV3Sb5 nanoflake tip after repeating the “injection process” for many times at several clean as-cleaved surface regions.”

Fig. R1. Temperature-dependent dI/dV (d^2I/dV^2) spectra obtained using the normal STM tip and the superconducting STM tip. a, A series of dI/dV spectra obtained by normal STM tip under different temperatures, showing the superconducting gaps and the bosonic mode gradually disappear with increasing temperature, and are invisible when electron temperature $T_{\text{electron}} > 2.06$ K ($V_s = -2.0$ mV, $V_{\text{mod}} = 0.1$ mV, $I_t = 1$ nA). b, A series of derivative spectra (d^2I/dV^2) obtained from a. c, A series of dI/dV spectra obtained by superconducting STM tip under different temperatures, showing clearly the signal of Josephson peak, superconducting coherence peak, and bosonic mode, and those signals are weakening with increasing temperature, giving a phase transition temperature of $T_{\text{electron}} \sim 2.00$ K ($V_s = -2.0$ mV, $V_{\text{mod}} = 0.1$ mV, $I_t = 600$ nA). d, A series of derivative spectra obtained from c. Superconducting peaks, bosonic mode, and the Josephson peak in a, b, c, and d are labeled and marked by the dashed lines.

I am grateful to the authors for the Josephson Tunneling measurements. The authors made a superconducting tip by picking up a CsV3Sb5 flake from the sample such that the spectral signatures on the tip are virtually identical to those on the sample. In the Josephson tunneling, I expect to see two different signatures – one is a conventional Josephson tunneling (“intra-band” Josephson tunneling between the same bands in the sample and tip), and another would be associated with a Leggett mode (“inter-band” Josephson tunneling between the different bands in the sample and tip). Thus, I expect to see a conventional Josephson branch in the tunneling near the zero bias and an enhancement of the current at $|V_{\text{bias}}| > \Omega$ due to Leggett mode. These signatures would show up as peaks near the zero bias and $|V_{\text{bias}}| \sim \Omega$ in dI/dV , respectively. Contributions from a single particle tunneling will be on top of them. I also expect to see a sudden change in these signatures at the temperature where a smaller gap in one band is closed. At this temperature, a Josephson Tunneling associated with the Leggett mode would disappear as well as the “intra-band” Josephson for the smaller gap, so that the corresponding spectral signatures would disappear below T_c .

In the data presented in Fig. R1, I don't see clear evidence of the Josephson Tunneling associated with the Leggett mode. The authors just observed the conventional Josephson current in Fig. R1.

I appreciate the authors for responses to my comments. I am also grateful to the authors for additional Josephson Tunneling measurements in the responses. While Fig. R1c captures the signature of the Josephson Tunneling, this looks just like a conventional Josephson Tunneling, and I don't see a clear signature associated with the Leggett mode in the data. Please see details below. I have to say that the manuscript lacks experimental evidence for the presence of the Leggett mode. I don't think that the peak-dip-hump structure in the local density of states can be evidence of the Leggett mode in the present study. Thus, I can't recommend for a publication in the Nature Communications. My responses are in green.

Comment 3: *The authors just analyzed ~300 dI/dV spectra for each x . First, it's not clear where these spectra are taken exactly in the field of view. Second and more importantly, it's not trivial that ~300 spectra are statistically enough to account for variations in the spectral shape (or heterogeneity in the electronic structure). It's possible that these spectra may not be representative but may be selected from local regions that is not an intrinsic property in this compound. Obviously, there are variations in the spectral shape as shown in Fig. 2 and 3, and the authors should demonstrate that ~300 spectra are enough. For example, one can take a dI/dV map in 50nm FOV with 256×256 pixels at given energy and construct a histogram of the values in dI/dV map. Then, one can simply crop/divide this map into four regions ($4 \times 25 \text{ nm}^2$ 128×128 pixels maps) and construct the histogram again for each map. If the size of the FOV and pixels# are statistically enough, then I would expect that widths of the distribution are the same before and after the crop. Alternatively, and more convincingly, the authors can take a data in the same way as in the Ref. 2, constructing both Δ and Ω maps and visualizing spatial structure of them.*

Response 3: We thank the reviewer for the suggestion on constructing $\Delta(\mathbf{r})$ and $\Omega(\mathbf{r})$ maps. **New Extended Data Fig.2 (Fig. R3)** in the revised manuscript shows the result of $\Delta(\mathbf{r})$ and $\Omega(\mathbf{r})$ collected within a field of view of $25 \times 25 \text{ nm}$ and 180×180 pixels. The spatial variation and fertile patterns are visualized clearly, implying the spatial variation of dI/dV spectra in CsV_3Sb_5 .

Fig. R3. Spatial distribution of $\Delta(\mathbf{r})$ and $\Omega(\mathbf{r})$ of CsV_3Sb_5 . a, Spatial distribution of $\Delta(\mathbf{r})$

obtained on the Sb surface within the field of view of $25 \text{ nm} \times 25 \text{ nm}$, showing the spatial inhomogeneity of superconducting gap ($V_s=-2.0 \text{ mV}$, $I_r=1 \text{ nA}$, $V_{mod}=0.1 \text{ mV}$). **b**, The histogram plot of **a**. **c**, Spatial distribution of $\Omega(\mathbf{r})$ obtained on the same field of view of **a**, showing the spatial inhomogeneity of the bosonic mode. **d**, The histogram plot of **c**.

Accordingly, we added spatial distribution of $\Delta(\mathbf{r})$ and $\Omega(\mathbf{r})$ as new Extended Data Fig. 2 and revised the statement to clarify the methodology in the revised manuscript, as follows:

“In Fig. 2c, we show a series of dI/dV curves with the dominant one-gap line-shape in a stackplot without offset from different regions and samples. The spatial distribution of $\Delta(\mathbf{r})$ and $\Omega(\mathbf{r})$ within a field of view of $25 \text{ nm} \times 25 \text{ nm}$ are presented in Extended Data Fig. 2.”

I understand the statistics. But it looks to me that there is a tip change in the data, and data quality at the top 25% of the FOV is quite different where the gap values are highly uniform around 0.35 meV . This would largely contribute to the histogram in Fig. R3b. Nevertheless, a full width the half maximum is about 0.02 meV that is about 6% from its average. This suggests that the Δ is highly uniform compared to other compounds such as those in cuprates.

Comment 6: *A Leggett mode energy is proportional to the superconducting order parameter amplitude (S.G. Sharapov et al., Eur. Phys. J. B 30, 45–51 (2002)). Thus, I would expect that the mode energy Ω should increase with increasing the gap size Δ , unless a coupling constant changes dramatically with x . However, as shown in Fig. 4, Ω and Δ show an opposite trend - Ω decreases but Δ increases with x . Why do they show the opposite trend with x ?*

Response 6: We thank the reviewer for this critical comment. In the primary work studied by Professor A. J. Leggett (Prog. Theor. Phys. **36**, 901-930 (1966)), the mode energy expressed as:

$$E \propto 4 \sqrt{\frac{1}{2} \left(\frac{1}{\rho_1} + \frac{1}{\rho_2} \right) \left(\frac{J}{UV} \right) \Delta_1 \Delta_2}$$

where $\rho_{1,2}$ is the density of states at Fermi surface, U and V are the pairing potential, J is the Josephson coupling between the two superconducting components, $\Delta_{1,2}$ is the superconducting order parameter. It is natural to expect a positive correlation between the superconducting gap (or T_c) and mode energy when assuming an unchanged density of states. However, with the introduction of Ta substitution, the normal state density will get an enhancement, stemming from the weakening of the charge density wave. When the change in the normal state density is much significant than in the superconducting gap, an anti-correlation between the superconducting gap and bosonic mode energy emerges.

It should be also noted that the anti-correlation between superconducting gap and Leggett mode energy has been well-studied in single-layer NbSe_2 (*Adv. Mater.* **34**, e2206078 (2022)).

To avoid confusion on the anti-correlation between the superconducting gap and Leggett mode energy emerges in the Kagome superconductors, we have added discussions in the revised manuscript as follows:

“We next turn to discuss the possible origin of the bosonic mode. In Fig. 4c, the mode energy Ω is plotted as a function of the Ta-substitution x . It decreases continuously with increasing x accompanied by the increase of the SC gap Δ defined by the peak-to-peak distance in the conductance spectra and the suppression of the CDW order (Fig. 3). **Such an anti-correlation between the Leggett mode and the**

superconducting gap has also been observed in monolayer NbSe₂¹⁴. The Leggett mode is expected to be proportional to the SC gaps but anti-correlated with the density of states¹². When the CDW order in kagome superconductors is suppressed, the normal state density is likely to gain a more significant enhancement than the superconducting gap, resulting in an anti-correlation relationship.

I am not convinced by these arguments. In general, in the CDW ordered phase, the Fermi surface is expected to be reconstructed and be (partially) gapped out. ρ_1 and ρ_2 are the density of states at the Fermi level, so they are supposed to be (partially) gapped out in the CDW ordered phase. So, I don't understand why ρ_1 and ρ_2 are enhanced by weakening the CDW states. ρ_1 and ρ_2 are expected to be independent on the magnitude of the CDW order parameter.

Comment 8: *In line 276, 'The low-energy Raman modes observed in CsV₃Sb₅ is at least 5.5 meV in energy^{60,61}, which is well above the bosonic mode energy. We can safely rule out phonons as an origin since there has been no observed phonon modes at such low energies.'* In fact, Ref. 61 reported dispersions of the Longitudinal and Transverse acoustic phonons. I don't think that the author can rule out a possibility that one of these phonons are coupled to the electrons, such that the Bogoliubov quasiparticle dispersion is renormalized by the electronphonon coupling, resulting in a peak-dip-hump structure in dI/dV spectra.

Response 8: We thank the reviewer for the thoughtful consideration. Despite the dispersive nature of longitudinal and transverse acoustic phonons, the spectral weight is notably concentrated at ~ 5.5 meV (*Phys. Rev. X* **11**, 031050 (2021)). Consequently, we do not anticipate these phonons to effectively couple with electrons. In addition, the bosonic mode observed in our manuscript can exist when the CDW is totally suppressed by Ta substitution, thus we exclude its phonon origin. We also stress the argument in the manuscript as follows:

"The spectral weight of the Raman mode observed in CsV₃Sb₅ peaks at 5.5 meV in energy^{65,66}, which is well above the bosonic mode energy. We can safely rule out phonons as an origin since there has been no observed phonon modes at such low energies."

Raman can't measure a dispersion of the phonons. The energy measured by Raman would be a value near the zone edge in momentum transfer space or at a momentum transfer constrained by the symmetry in the measurement. A phonon that may couple to electrons can be at finite momentum transfer at different energy from 5.5meV, depending on the band structure and the phonon dispersion. I don't think that the authors can rule out a possibility of the phonon origin.

Comment 9: *Regarding the Fig. 4f, Ref. 2 reported that the peak-dip-hump structure is due to the electron-phonon coupling. However, there is virtually no change in T_c when ¹⁶O is substituted by ¹⁸O, while the lattice vibration frequency is shifted by the amount of a difference in the oxygen mass in Bi2212. It's also the fact that the Ω is doping independent from slightly underdoped to overdoped regime, while Δ decreases with increasing doping as reported in the Ref. 2. Bi2212 obviously deviates from the line in Fig. 4f.*

Response 9: Yes, we have taken the data from *Nature* **442**, 546-550 (2006) into Fig. 4f without introducing any bias. We have refrained from commenting on the behaviour of Ω in cuprate in our manuscript.

Nature **442**, 546-550 (2006) eventually studied the doping dependence in addition to the isotope effect. It's strange to selectively pick up values only for the O18 and O16 samples in Fig. 4f. The authors should include the Ω values for different doping levels from this reference as the authors mention 'without introducing any bias'.

Comment 11: *I don't understand why the authors divided the data points in Fig. 4f into two groups as in the figure. I can understand if these data points are grouped by those that follow the linear relation and those that deviate from the line. In fact, Bi2212 also deviates from this line, so I don't think the Bi2212 can be the same group as other high T_c superconductors in this plot. The manuscript is not much benefitted from the Fig. 4f.*

Response 11: We express our gratitude to the reviewer for their thoughtful consideration. We believe that Fig. 4f in our manuscript provides the most informative representation and is plotted without bias. Notably, we abstained from attempting to cluster or group the data points, and the inclusion of the Uemura line is merely intended as a guiding reference. It is crucial to emphasize that we refrained from drawing additional conclusions based on Fig. 4f, despite the divergence of kagome superconductors, moiré materials, and cuprates from the dashed Uemura line. In the revised manuscript, we amended the statement regarding the high- T_c superconductors as follows:

“For most of high- T_c superconductors, the STM bosonic mode follows the Uemura line⁵ ($k_B T_c \sim \Omega/4$). While the stoichiometric CsV₃Sb₅ still lies close to this line, as does NbSe₂ film, Ta substitution systematically drives the ratio $\Omega/k_B T_c$ away from the Uemura value. This is likely due to the nature of the low-lying SC collective excitations.”

Obviously, as I pointed out, T_c - Ω relationship in Bi2212 strongly deviates from the dashed line in Fig. 4, although the authors didn't include the T_c - Ω values for different doping levels from the ref. 2 (J. Lee et al., *Nature* **442**, 546-550 (2006)). Cuprates are effectively a single-band system, and no Leggett mode has been realized. Thus, the deviation from the dashed line doesn't necessarily imply an existence of the SC collective excitations.

Comment 13: *I would suggest the authors to perform Josephson Tunneling Microscopy measurement using the superconducting tip. I would expect a feature associated with the Leggett mode in the Josephson current spectra if exists.*

Response 13: We thank the reviewer for the insightful suggestion. We have constructed the Josephson tunnelling spectroscopy and investigated the bosonic mode further. Fig. R1 (New Extended Data Fig. 3) shows that the signal of bosonic mode becomes more significant by using the Josephson tunnelling spectroscopy, and the bosonic mode accompanied by the superconducting gap and Josephson current disappears when $T > T_c$, demonstrating a superconducting collective mode.

In the revised manuscript, we add the description on fabrication of superconducting STM tip in the method part as follows:

“**Fabrication of the superconducting CsV₃Sb₅ nanoflake tip.** The tungsten STM tip was used to inject the clean surface of the sample more than 15 nm depth, holding for 5~10 s with the voltage 1~2 V, and withdraw to its original position. After that, the CsV₃Sb₅ nanoflake will stick to the apex of the tungsten tip. We usually obtained the stable CsV₃Sb₅ nanoflake tip after repeating the “injection process” for many times at several clean as-cleaved surface regions.”

Fig. R1. Temperature-dependent dI/dV (d^2I/dV^2) spectra obtained using the normal STM tip and the superconducting STM tip. a, A series of dI/dV spectra obtained by normal STM tip under different temperatures, showing the superconducting gaps and the bosonic mode gradually disappear with increasing temperature, and are invisible when electron temperature $T_{\text{electron}} > 2.06$ K ($V_s = -2.0$ mV, $V_{\text{mod}} = 0.1$ mV, $I_t = 1$ nA). **b**, A series of derivative spectra (d^2I/dV^2) obtained from **a**. **c**, A series of dI/dV spectra obtained by superconducting STM tip under different temperatures, showing clearly the signal of Josephson peak, superconducting coherence peak, and bosonic mode, and those signals are weakening with increasing temperature, giving a phase transition temperature of $T_{\text{electron}} \sim 2.00$ K ($V_s = -2.0$ mV, $V_{\text{mod}} = 0.1$ mV, $I_t = 600$ nA). **d**, A series of derivative spectra obtained from **c**. Superconducting peaks, bosonic mode, and the Josephson peak in **a**, **b**, **c**, and **d** are labeled and marked by the dashed lines.

I am grateful to the authors for the Josephson Tunneling measurements. The authors made a superconducting tip by picking up a CsV_3Sb_5 flake from the sample such that the spectral signatures on the tip are virtually identical to those on the sample. In the Josephson tunneling, I expect to see two different signatures – one is a conventional Josephson tunneling (“intra-band” Josephson tunneling between the same bands in the sample and tip), and another would be associated with a Leggett mode (“inter-band” Josephson tunneling between the different bands in the sample and tip). Thus, I expect to see a conventional Josephson branch in the tunneling near the zero bias and an enhancement of the current at $|V_{\text{bias}}| > \Omega$ due to Leggett mode. These signatures would show up as peaks near the zero bias and $|V_{\text{bias}}| \sim \Omega$ in dI/dV , respectively. Contributions from a single particle tunneling will be on top of them. I also expect to see a sudden change in these signatures at the temperature where a smaller gap in one band is closed. At this temperature, a Josephson Tunneling associated with the Leggett mode would disappear as well as the “intra-band” Josephson for the smaller gap, so that the corresponding spectral signatures would disappear below T_c .

In the data presented in Fig. R1, I don’t see clear evidence of the Josephson Tunneling associated with the Leggett mode. The authors just observed the conventional Josephson current in Fig. R1.

Responses to the comments from the reviewers

Response to the Reviewer #1

In the revised version of the manuscript, the authors have addressed the issues that I pointed out in the previous report. Especially, they included additional data (Supplementary Figure 3) showing the temperature dependence of dI/dV spectra with both normal and superconducting STM tips. It is clearly seen that the peak-dip-hump structure disappears for $T > T_c$, indicating that it is correlated with the superconducting order. These data are very encouraging and interesting. The anti-correlation between the mode energy and doping level is also addressed from the point of view of density of states. This could be a possible explanation. In the reply to the referee report, it is also clearly explained how to determine E^\pm from the dI/dV curve.

We sincerely thank the reviewer for their positive comments.

Comment 1: How does the bosonic mode energy change with temperature? Does it approach zero as T gets closer to T_c (as expected in the case of Leggett mode)?

Response 1: We thank the reviewer for the question. Using the CsV_3Sb_5 nanoflake tip as shown in Fig. R1 (new Supplementary Fig. 3), from which we can see that the bosonic mode in energy decreasing and approaching to zero with temperature.

Fig. R1. Temperature-dependent dI/dV (d^2I/dV^2) spectra obtained using the superconducting STM tip. a, A series of dI/dV spectra obtained by superconducting STM tip under different temperatures, showing clearly the signal of Josephson peak, superconducting coherence peak, and bosonic mode, and those signals are weakening with increasing temperature, giving a phase transition temperature of $T_{\text{electron}} \sim 2.00$ K ($V_s = -2.0$ mV, $V_{\text{mod}} = 0.1$ mV, $I_t = 600$ nA). **b,** A series of derivative spectra obtained from **a**. The blue dashed line in **a** marks

the superconducting peaks and the Josephson peak. The black dashed lines in **b** mark the evolution of Δ and $|\Delta + \Omega|$.

In the revised manuscript, we add the black dashed lines to Supplementary Fig. 3.

Comment 2: *In the revised manuscript, it says that “Those dips/peaks are chosen by prominent, closest to the superconducting gaps, and particle-hole symmetry”, but this sentence sounds unclear. Can the authors improve this sentence? The sentence in the report is much better: “we choose the most prominent and symmetric local minimal/maximal that is closest to the coherence peak, while the other peaks/dips may be the data noise due to the weakness and particle particle-hole asymmetry”.*

Response 2: We agree with reviewer on the revision. Accordingly, in the revised manuscript, we replace the statement “Those dips/peaks are chosen by prominent, closest to the superconducting gaps, and particle-hole symmetry” to “We choose the most prominent and symmetric local minimal/maximal that is closest to the coherence peak, while the other peaks/dips may be the data noise due to the weakness and particle particle-hole asymmetry”.

Summary comment: *To summarize, the authors’ response to my previous comments is quite satisfactory. The paper is significantly improved, and positive support for the superconducting collective mode is obtained from the additional data. Based on these, I would like to recommend the publication of the paper in Nature Communications.*

We sincerely thank the reviewer for their positive comments.

Response to the Reviewer #2

Since the previous version, the authors have significantly improved their paper by adding new temperature dependence data, tunnelling spectra with superconducting tip and histograms for gap energy and mode energy in larger FOV. They also properly weaken the claim about gamma in the Dynes fittings. I think the new data make the conclusion more solid, which adds great value to this paper. Now, the paper provides at least a self-consistent observation of Leggett mode in this kagome superconductor family. I can support its publication in Nature Communications.

We sincerely thank the reviewer for their positive comments.

Response to the Reviewer #3

I appreciate the authors for responses to my comments. I am also grateful to the authors for additional Josephson Tunneling measurements in the responses. While Fig. R1a captures the signature of the Josephson Tunneling, this looks just like a conventional Josephson Tunneling, and I don't see a clear signature associated with the Leggett mode in the data. Please see details below. I have to say that the manuscript lacks experimental evidence for the presence of the Leggett mode. I don't think that the peak-dip-hump structure in the local density of states can be evidence of the Leggett mode in the present study. Thus, I can't recommend for a publication in the Nature Communications. My responses are in green.

We thank the reviewer for continuing to provide constructive criticisms, comments, and suggestions. We have conducted further analyses and experiments to address the reviewer's concerns. We hope that these inputs will convince the reviewer that the peak-dip-hump structures is associated with Leggett mode, making it suitable for publication in Nature Communications.

In the following, for convenience of reviewer, reviewer's previous comments are in *black italic* typeface, our previous responses are in dark blue, the previous changes to the text are in gold. In this round, our responses are in regular blue typeface and our changes to the text are in red.

Comment 1: *The authors just analyzed ~300 dI/dV spectra for each x . First, it's not clear where these spectra are taken exactly in the field of view. Second and more importantly, it's not trivial that ~300 spectra are statistically enough to account for variations in the spectral shape (or heterogeneity in the electronic structure). It's possible that these spectra may not be representative but may be selected from local regions that is not an intrinsic property in this compound. Obviously, there are variations in the spectral shape as shown in Fig. 2 and 3, and the authors should demonstrate that ~300 spectra are enough. For example, one can take a dI/dV map in 50nm FOV with 256×256 pixels at given energy and construct a histogram of the values in dI/dV map. Then, one can simply crop/divide this map into four regions ($4 \times 25 \text{ nm}^2$ - 128×128 pixels maps) and construct the histogram again for each map. If the size of the FOV and pixels# are statistically enough, then I would expect that widths of the distribution are the same before and after the crop. Alternatively, and more convincingly, the authors can take a data in the same way as in the Ref. 2, constructing both Δ and Ω maps and visualizing spatial structure of them.*

Response 1: We thank the reviewer for the suggestion on constructing $\Delta(\mathbf{r})$ and $\Omega(\mathbf{r})$ maps. Fig.R2 in the revised manuscript shows the result of $\Delta(\mathbf{r})$ and $\Omega(\mathbf{r})$ collected within a field of view of $25 \times 25 \text{ nm}$ and 180×180 pixels. The spatial variation and fertile patterns are visualized clearly, implying the spatial variation of dI/dV spectra in CsV_3Sb_5 .

Fig. R2. Spatial distribution of $\Delta(\mathbf{r})$ and $\Omega(\mathbf{r})$ of CsV_3Sb_5 . **a**, Spatially distribution of $\Delta(\mathbf{r})$ obtained on the Sb surface within the field of view of $25 \text{ nm} \times 25 \text{ nm}$, showing the spatially inhomogeneity of superconducting gap ($V_s = -2.0 \text{ mV}$, $I_r = 1 \text{ nA}$, $V_{mod} = 0.1 \text{ mV}$). **b**, The histogram plot of **a**. **c**, Spatial distribution of $\Omega(\mathbf{r})$ obtained on the same field of view of **a**, showing the spatial inhomogeneity of the bosonic mode. **d**, The histogram plot of **c**.

Accordingly, we added spatial distribution of $\Delta(\mathbf{r})$ and $\Omega(\mathbf{r})$ as new Extended Data Fig. 2 and revised the statement to clarify the methodology in the revised manuscript, as follows:

“In Fig. 2c, we show a series of dI/dV curves with the dominant one-gap line-shape in a stack-plot without offset from different regions and samples. The spatial distribution of $\Delta(\mathbf{r})$ and $\Omega(\mathbf{r})$ within a field of view of $25 \text{ nm} \times 25 \text{ nm}$ are presented in Extended Data Fig. 2.”

New Comment 1: I understand the statistics. But it looks to me that there is a tip change in the data, and data quality at the top 25% of the FOV is quite different where the gap values are highly uniform around 0.35 meV . This would largely contribute to the histogram in Fig. R1b. Nevertheless, a full width the half maximum is about 0.02 meV that is about 6% from its average. This suggests that the Δ is highly uniform compared to other compounds such as those in cuprates.

Response 1: We thank the reviewer for bringing up this question. As the reviewer has pointed out, it is important to check if the tip change in the $\Delta(\mathbf{r})$. The three aspects on topography $T(\mathbf{r})$, dI/dV spectra, and dI/dV maps are as follow:

1. Figure R2_1 **a** is the simultaneously obtained $T(\mathbf{r})$ of $\Delta(\mathbf{r})$, which shows the tip condition is preserved during the measurement.
2. Figure R2_1 **b** is the stack plot of dI/dV spectra with a step of 5 pixels as in $\Delta(\mathbf{r})$, which

shows the superconducting gap varies smoothly within the statistical histogram during the measurement.

- Figure R2_1 **c-f** are the typical dI/dV maps in energy range from 0.3 meV to 0.8 meV in line with the histogram in Fig. R2b, which show the preserved tip condition during the measurement.

Fig. R2_1. Topography, STS and dI/dV maps. **a**, Topography $T(\mathbf{r})$ obtained simultaneously and in the same field of view of $\Delta(\mathbf{r})$. **b**, A series of dI/dV spectra obtained simultaneously and in the same field of view of **a**, which generates the $\Delta(\mathbf{r})$. **c-f**, A series of dI/dV maps obtained at energy of -0.333 meV (**c**), -0.347 meV (**d**), -0.360 meV (**e**), -0.373 meV (**f**).

It also should be noticed that the simultaneously obtained $\Omega(\mathbf{r})$ in Fig. R2c do not show the signature regarding the tip change. Although the superconducting gap size varies in a small field of view, the inhomogeneity of the superconductor lays on the coherence peaks, in-gap conductance, and the Bogoliubov quasiparticle density of states beyond the superconducting gap, especially compared to the Ta-substituted components.

On the other hand, we conduct new $\Delta(\mathbf{r})$ map within a field view of $15 \text{ nm} \times 15 \text{ nm}$ as shown in Fig. R2_2 (new Supplementary Figure 2). The full width of the half maximum is about 0.02 meV as well.

Fig. R2_2 (new Supplementary Figure 2). Spatial distribution of $\Delta(\mathbf{r})$ and $\Omega(\mathbf{r})$ of CsV_3Sb_5 . **a**, Spatially distribution of $\Delta(\mathbf{r})$ obtained on the Sb surface within the field of view of $15 \text{ nm} \times 15 \text{ nm}$, showing the spatial inhomogeneity of superconducting gap ($V_s = -2.0 \text{ mV}$, $I_f = 1 \text{ nA}$, $V_{mod} = 0.1 \text{ mV}$). **b**, The histogram plot of **a**. **c**, Spatial distribution of $\Omega(\mathbf{r})$ obtained on the same field of view of **a**, showing the spatial inhomogeneity of the bosonic mode. **d**, The histogram plot of **c**.

Comment 2: A Leggett mode energy is proportional to the superconducting order parameter amplitude (S.G. Sharapov et al., *Eur. Phys. J. B* 30, 45–51 (2002)). Thus, I would expect that the mode energy Ω should increase with increasing the gap size Δ , unless a coupling constant changes dramatically with x . However, as shown in Fig. 4, Ω and Δ show an opposite trend - Ω decreases but Δ increases with x . Why do they show the opposite trend with x ?

Response 2: We thank the reviewer for this critical comment. In the primary work studied by Professor A. J. Leggett (*Prog. Theor. Phys.* 36, 901-930 (1966)), the mode energy expressed as:

$$E \propto 4 \sqrt{\frac{1}{2} \left(\frac{1}{\rho_1} + \frac{1}{\rho_2} \right) \left(\frac{J}{UV} \right) \Delta_1 \Delta_2}$$

where $\rho_{1,2}$ is the density of states at Fermi surface, U and V are the pairing potential, J is the Josephson coupling between the two superconducting components, $\Delta_{1,2}$ is the superconducting order parameter. It is natural to expect a positive correlation between the superconducting gap (or T_c) and mode energy when assuming an unchanged density of states.

However, with the introduction of Ta substitution, the normal state density will get an enhancement, stemming from the weakening of the charge density wave. When the change in the normal state density is much significant than in the superconducting gap, an anti-correlation between the superconducting gap and bosonic mode energy emerges.

It should be also noted that the anti-correlation between superconducting gap and Leggett mode energy has been well-studied in single-layer NbSe₂ (*Adv. Mater.* **34**, e2206078 (2022)).

To avoid confusion on the anti-correlation between the superconducting gap and Leggett mode energy emerges in the Kagome superconductors, we have added discussions in the revised manuscript as follows:

“We next turn to discuss the possible origin of the bosonic mode. In Fig. 4c, the mode energy Ω is plotted as a function of the Ta-substitution x . It decreases continuously with increasing x accompanied by the increase of the SC gap Δ defined by the peak-to-peak distance in the conductance spectra and the suppression of the CDW order (Fig. 3). Such an anti-correlation between the Leggett mode and the superconducting gap has also been observed in monolayer NbSe₂¹⁴. The Leggett mode is expected to be proportional to the SC gaps but anti-correlated with the density of states¹². When the CDW order in kagome superconductors is suppressed, the normal state density is likely to gain a more significant enhancement than the superconducting gap, resulting in an anti-correlation relationship.”

New Comment 2: I am not convinced by these arguments. In general, in the CDW ordered phase, the Fermi surface is expected to be reconstructed and be (partially) gapped out. ρ_1 and ρ_2 are the density of states at the Fermi level, so they are supposed to be (partially) gapped out in the CDW ordered phase. So, I don't understand why ρ_1 and ρ_2 are enhanced by weakening the CDW states. ρ_1 and ρ_2 are expected to be independent on the magnitude of the CDW order parameter.

Response 2: We thank the reviewer for the insights regarding the Fermi surface and the density of states in the presence of CDW. For both superconductivity and CDW, the density of states at the Fermi level plays an essential role (*npj Quantum Materials* **5**, 22 (2020), *Supercond. Sci. Technol.* **14**, R1–R27 (2001)). The reviewer correctly points out that the density of states at the Fermi level could be reduced in the CDW ordered phase. In turn, suppression of the CDW by Ta substitution begins to close the CDW gap, leading to the release of the previously gapped Fermi level density of states. An increase in ρ_1 and ρ_2 would then be expected. Furthermore, our recent ARPES work report the substantially enhanced density of states at the Fermi level induced by the Ta doping (Supplementary Fig. S19 in *Nat. Commun.* **14**, 3819 (2023))

Comment 3: In line 276, ‘The low-energy Raman modes observed in CsV₃Sb₅ is at least 5.5 meV in energy^{60,61}, which is well above the bosonic mode energy. We can safely rule out phonons as an origin since there has been no observed phonon modes at such low energies.’.

In fact, Ref. 61 reported dispersions of the Longitudinal and Transverse acoustic phonons. I don't think that the author can rule out a possibility that one of these phonons are coupled to the electrons, such that the Bogoliubov quasiparticle dispersion is renormalized by the electron-phonon coupling, resulting in a peak-dip-hump structure in dI/dV spectra.

Response 3: We thank the reviewer for the thoughtful consideration. Despite the dispersive nature of longitudinal and transverse acoustic phonons, the spectral weight is notably concentrated at ~ 5.5 meV (*Phys. Rev. X* **11**, 031050 (2021)). Consequently, we do not anticipate these phonons to effectively couple with electrons. In addition, the bosonic mode observed in our manuscript can exist when the CDW is totally suppressed by Ta substitution, thus we exclude its phonon origin. We also stress the argument in the manuscript as follows:

“The spectral weight of the Raman mode observed in CsV_3Sb_5 peaks at 5.5 meV in energy^{65,66}, which is well above the bosonic mode energy. We can safely rule out phonons as an origin since there has been no observed phonon modes at such low energies.”

New Comment 3: Raman can't measure a dispersion of the phonons. The energy measured by Raman would be a value near the zone edge in momentum transfer space or at a momentum transfer constrained by the symmetry in the measurement. A phonon that may couple to electrons can be at finite momentum transfer at different energy from 5.5meV, depending on the band structure and the phonon dispersion. I don't think that the authors can rule out a possibility of the phonon origin.

Response 3: We thank the Reviewer for this comment. We agree with the Reviewer that Raman cannot measure the dispersion of phonons and the phonon theoretically could couple to the electrons at finite momentum transfer at small energies. These high-order phonon modes are expected to be much weaker than the phonon modes corresponding to Raman measurements. We have tried the low-energy Raman measurements but are unable to resolve the signature within the energy of this mode. In addition, there have been no observations on such high-order phonon mode at lower energy than Raman mode in previous STM works on superconductors as well. Even so, we cannot completely rule out low-energy Raman mode when phonon contributes to the dI/dV spectra.

In the revised manuscript, we tone down our statement on the phonon mode:

“The spectral weight of the Raman mode observed in CsV_3Sb_5 peaks at 5.5 meV in energy^{65,66}, which is well above the bosonic mode energy. We **approximately** rule out phonons as an origin **since there has been no observed phonon modes at much lower energies than Raman mode in previous STM works on superconductors.**”

Comment 4: Regarding the Fig. 4f, Ref. 2 reported that the peak-dip-hump structure is due to the electron-phonon coupling. However, there is virtually no change in T_c when ^{16}O is substituted by ^{18}O , while the lattice vibration frequency is shifted by the amount of a difference

in the oxygen mass in Bi2212. It's also the fact that the Ω is doping independent from slightly underdoped to overdoped regime, while Δ decreases with increasing doping as reported in the Ref. 2. Bi2212 obviously deviates from the line in Fig. 4f.

Response 4: Yes, we have taken the data from *Nature* 442, 546-550 (2006) into Fig. 4f without introducing any bias. We have refrained from commenting on the behaviour of Ω in cuprate in our manuscript.

New Comment 4: *Nature* 442, 546-550 (2006) eventually studied the doping dependence in addition to the isotope effect. It's strange to selectively pick up values only for the O^{18} and O^{16} samples in Fig. 4f. The authors should include the Ω values for different doping levels from this reference as the authors mention 'without introducing any bias'.

Response 4: We thank the reviewer for the constructive comment. We include the $\Omega - T_c$ relation in Fig. R3. The T_c s for different doping levels of BSCCO are taken from *Nat. Commun.* 9, 5210 (2018). Since the Ω values do not change with doping, which indicates a lattice mode as referred *Nature* 442, 546-550 (2006), we integrate the doping effect into the error bar of T_c .

Fig. R3, Plot of transition temperature T_c as a function of mode energy Ω . The data are taken or extracted from mostly STM works on kagome superconductors reported here, $NbSe_2$ ¹⁴, twisted bilayer graphene⁵⁷, twisted trilayer graphene⁵⁸, $Pr_{0.88}LaCe_{0.12}CuO_4$ ³, $Bi_2Sr_2CaCu_2O_{8+\delta}$ ², one-unit-cell $FeSe$ ⁴, $EuRbFe_4As_4$ ⁵⁹, $Ba_{0.6}K_{0.4}Fe_2As_2$ ^{46,47}, and $Na(Fe_{0.975}Co_{0.025})As$ ⁴⁶, as well as from the Uemura plot⁵. The T_c s of twisted trilayer graphene are extracted from Ref⁶⁴, the T_c s of BSCCO are extracted from *Nat. Commun.* 9, 5210 (2018).

To avoid the potential misunderstanding on the judgement of the SC collective mode, we have excluded the BSCCO results from Fig. 4f, where the new Fig. 4f is shown as the Fig. R4 (please see the response to Comment 5).

Comment 5: I don't understand why the authors divided the data points in Fig. 4f into two groups as in the figure. I can understand if these data points are grouped by those that follow

the linear relation and those that deviate from the line. In fact, Bi2212 also deviates from this line, so I don't think the Bi2212 can be the same group as other high T_c superconductors in this plot. The manuscript is not much benefitted from the Fig. 4f.

Response 5: We express our gratitude to the reviewer for their thoughtful consideration. We believe that Fig. 4f in our manuscript provides the most informative representation and is plotted without bias. Notably, we abstained from attempting to cluster or group the data points, and the inclusion of the Uemura line is merely intended as a guiding reference. It is crucial to emphasize that we refrained from drawing additional conclusions based on Fig. 4f, despite the divergence of kagome superconductors, moiré materials, and cuprates from the dashed Uemura line.

In the revised manuscript, we amended the statement regarding the high- T_c superconductors as follows:

“For most of high- T_c superconductors, the STM bosonic mode follows the Uemura line⁵ ($k_B T_c \sim \Omega/4$). While the stoichiometric CsV_3Sb_5 still lies close to this line, as does NbSe_2 film, Ta substitution systematically drives the ratio $\Omega/k_B T_c$ away from the Uemura value. This is likely due to the nature of the low-lying SC collective excitations.”

New Comment 5: Obviously, as I pointed out, $T_c - \Omega$ relationship in Bi2212 strongly deviates from the dashed line in Fig.4, although the authors didn't include the $T_c - \Omega$ values for different doping levels from the ref. 2 (J. Lee et al., Nature 442, 546-550 (2006)). Cuprates are effectively a single-band system, and no Leggett mode has been realized. Thus, the deviation from the dashed line doesn't necessarily imply an existence of the SC collective excitations.

Response 5: We thank the reviewer for the insightful consideration and point out the important comment. Collective excitation in Nature 442, 546-550 (2006) is most likely linked to the lattice. This mode deviate from the dashed line in Fig.4 may lead to misunderstanding on the judgement of SC collective excitation. In the revised manuscript, we have removed these data of BSCCO from Fig. 4f, and the new Fig. 4f is presented below (Fig. R4).

Fig. R4 (new Fig. 4f), Plot of transition temperature T_c as a function of mode energy Ω . The data are taken or extracted from mostly STM works on kagome superconductors reported here, NbSe_2 ¹⁴, twisted bilayer graphene⁵⁷, twisted trilayer graphene⁵⁸, $\text{Pr}_{0.88}\text{LaCe}_{0.12}\text{CuO}_4$ ³, $\text{Bi}_2\text{Sr}_2\text{CaCu}_2\text{O}_{8+\delta}$ ², one-unit-cell FeSe ⁴, $\text{EuRbFe}_4\text{As}_4$ ⁵⁹, $\text{Ba}_{0.6}\text{K}_{0.4}\text{Fe}_2\text{As}_2$ ^{46,47}, and $\text{Na}(\text{Fe}_{0.975}\text{Co}_{0.025})\text{As}$ ⁴⁶, as well as from the Uemura plot⁵. The T_c s of twisted trilayer graphene are extracted from Ref⁶⁴.

Comment 6: I would suggest the authors to perform Josephson Tunneling Microscopy measurement using the superconducting tip. I would expect a feature associated with the Leggett mode in the Josephson current spectra if exists.

Response 6: We thank the reviewer for the insightful suggestion. We have constructed the Josephson tunnelling spectroscopy and investigated the bosonic mode further. Fig. R5 shows that the signal of bosonic mode becomes more significant by using the Josephson tunnelling spectroscopy, and the bosonic mode accompanied by the superconducting gap and Josephson current disappears when $T > T_c$, demonstrating a superconducting collective mode.

In the revised manuscript, we add the description on fabrication of superconducting STM tip in the method part as follows:

“Fabrication of the superconducting CsV_3Sb_5 nanoflake tip. The tungsten STM tip was used to inject the clean surface of the sample more than 15 nm depth, holding for 5~10 s with the voltage 1~2 V, and withdraw to its original position. After that, the CsV_3Sb_5 nanoflake will stick to the apex of the tungsten tip. We usually obtained the stable CsV_3Sb_5 nanoflake tip after repeating the “injection process” for many times at several clean as-cleaved surface regions.”

Fig. R5. Temperature-dependent dI/dV (d^2I/dV^2) spectra obtained using the normal STM tip and the superconducting STM tip. a, A series of dI/dV spectra obtained by normal STM tip under different temperatures, showing the superconducting gaps and the bosonic mode gradually disappear with increasing temperature, and are invisible when electron temperature $T_{\text{electron}} > 2.06$ K ($V_s = -2.0$ mV, $V_{\text{mod}} = 0.1$ mV, $I_t = 1$ nA). b, A series of derivative spectra (d^2I/dV^2) obtained from a. c, A series of dI/dV spectra obtained by superconducting STM tip under different temperatures, showing clearly the signal of Josephson peak, superconducting coherence peak, and bosonic mode, and those signals are weakening with increasing temperature, giving a phase transition temperature of $T_{\text{electron}} \sim 2.00$ K ($V_s = -2.0$ mV, $V_{\text{mod}} = 0.1$ mV, $I_t = 600$ nA). d, A series of derivative spectra obtained from c. Superconducting peaks, bosonic mode, and the Josephson peak in a, b, c, and d are labeled and marked by the dashed lines.

New Comment 6: I am grateful to the authors for the Josephson tunneling measurements. The authors made a superconducting tip by picking up a CsV_3Sb_5 flake from the sample such that the spectral signatures on the tip are virtually identical to those on the sample. In the Josephson tunneling, I expect to see two different signatures – one is a conventional Josephson tunneling (“intra-band” Josephson tunneling between the same bands in the sample and tip), and another would be associated with a Leggett mode (“inter-band” Josephson tunneling between the different bands in the sample and tip). Thus, I expect to see a conventional Josephson branch

in the tunneling near the zero bias and an enhancement of the current at $|V_{bias}| > \Omega$ due to Leggett mode. These signatures would show up as peaks near the zero bias and $|V_{bias}| \sim \Omega$ in dI/dV , respectively. Contributions from a single particle tunneling will be on top of them. I also expect to see a sudden change in these signatures at the temperature where a smaller gap in one band is closed. At this temperature, a Josephson tunneling associated with the Leggett mode would disappear as well as the “intra-band” Josephson for the smaller gap, so that the corresponding spectral signatures would disappear below T_c . In the data presented in Fig. R5, I don’t see clear evidence of the Josephson tunneling associated with the Leggett mode. The authors just observed the conventional Josephson current in Fig. R5.

Response 6: We thank the reviewer for the insightful comments regarding the Josephson tunneling measurements. The reviewer is correct, inter-band and intra-band Josephson tunneling are both expected when using the CsV_3Sb_5 nanoflake tip (Fig. R5_1a). The intra-band Josephson tunneling is in analogy with the famous “Fujita conjecture” as well-studied in cuprate (*Nature* **580**, 65-70 (2020)).

It should be noticed that the signal at energy around $|\Omega|$ could merge into the coherence peak shoulder, then leading to a visually weak feature. Despite this challenge in experiments, we can still observe the peaks $|V_{bias}| \sim \Omega$ in dI/dV when the “inter-band” Josephson tunneling between the different bands in the sample and tip is dominated. As shown in Fig. R5_1 (new Supplementary Figure 4), the symmetric peaks tailing to the coherence peaks are labeled.

Fig. R5_1 (new Supplementary Figure 4). Two types of Josephson tunnelling and the signature of Leggett mode. **a**, Schematic of “intra-band” Josephson tunneling between the same band of sample and tip, and “inter-band” Josephson tunneling between different bands of sample and tip. **b**, The dI/dV spectrum obtained by the superconducting STM tip, showing the bosonic mode manifest itself at energy $\sim \Omega$ ($V_s = -10.0$ mV, $V_{mod} = 0.1$ mV, $I_t = 1200$ nA).

We thank the reviewer for the great suggestion on temperature-dependent Josephson tunneling measurements. Let us focus on Fig. R5_2 at first. Since the peaks at energy around $|\Omega|$ are weak and could merged into the convolved coherence peak, we mark the dips and peaks in the derivative dI/dV spectra in Fig. R5_2, where the collective mode manifested at energies around

$|\Omega|$ and $|\Delta + \Omega|$. With increasing temperature to ~ 1.86 K, the signal at energy around $|\Omega|$ disappears due to the disappearance of the smaller gap as pointed out by the reviewer, while the signal at energy around $|\Delta + \Omega|$ can continue to be observed.

Fig. R5_2. Temperature-dependent dI/dV (d^2I/dV^2) spectra obtained using the superconducting STM tip. a, A series of dI/dV spectra obtained by superconducting STM tip under different temperatures, showing clearly the signal of Josephson peak, superconducting coherence peak, and bosonic mode, and those signals are weakening with increasing temperature, giving a phase transition temperature of $T_{\text{electron}} \sim 2.00$ K ($V_s = -2.0$ mV, $V_{\text{mod}} = 0.1$ mV, $I_T = 600$ nA). **b,** A series of derivative spectra obtained from **a**. The blue dashed line in **a** mark the superconducting peaks and the Josephson peak. The black dashed lines in **b** mark the evolution of Δ and $|\Delta + \Omega|$.

However, it is not feasible for us to perform a deeper investigation of the Leggett mode evolution associated with “intra-band” and “inter-band” Josephson tunneling in experiments, due to the resolution of our instrument and the challenge of keeping the temperature at around 2 K for a long time. On the other hand, the Leggett mode could couple to the Higgs mode as introduced in the manuscript, which could also weaken the spectral signal at $|E| = \Omega$. Following the reviewer’s insightful suggestion, we can look at the pure Leggett in the feature.

Accordingly, we add a new method part to revised manuscript:

Discussion on the Leggett mode in Josephson tunneling spectroscopy. CsV_3Sb_5 is a two-band superconductor, so two types of Josephson tunneling, namely “intra-band” and “inter-band” Josephson tunneling (Supplementary Fig. 4a), will contribute to the experimental signal when the CsV_3Sb_5 nanoflake tip is used. This scenario is in analogy to the well-known “Fujita conjecture” proposed for cuprates. As shown in Supplementary Fig. 4a, the “intra-band” Josephson tunneling is between the same band in the sample and the tip, and the “inter-band” Josephson tunneling is between different bands in the sample and the tip. In this scenario, the Leggett mode is expected to enhance the tunneling current when the energy is greater than Ω , which is manifested as a peak in

the dI/dV spectrum. Supplementary Figure. 4b shows the signature of the Leggett mode at $|E| = \Omega$.

And the main text is revised according to Method:

In addition, the bosonic mode manifest itself at energy around Ω is also observed when using the superconducting STM tip, which is due to the potential “inter-band” Josephson tunneling (Method).

REVIEWERS' COMMENTS

Reviewer #1 (Remarks to the Author):

In the response to the second round of the reviewing process, the authors have addressed the comments of the reviewers in a fairly satisfactory manner. They added the data for the temperature dependence of the bosonic mode energy in Supplementary Fig. 3d, supporting the Leggett mode scenario (the phonon origin could be excluded). While it is not clear whether the observed signal comes from the intra-band or inter-band Josephson tunnelling, as pointed out by reviewer #3, at least the overall data presented in the paper seem to be consistent with the Leggett mode (inter-band Josephson mode). The ultimate identification may be open to future works, but the presented data are already interesting and worth to be published. I recommend to add a discussion on this point (the intra-band or inter-band mode) not only in the method section but also in the main text. With this, I can support the publication in Nature Communications.

Reviewer #2 (Remarks to the Author):

In the rebuttal letter, the authors response to the Reviewer's comments in details. There are basically three concerns by Reviewer 3. The first one is on the statistics of SC gap and Bosonic energy. In the letter, the authors have included more dI/dV maps and a new data set to support the claim; I find the experimental data being convincing to extract the gap energy and Bosonic energy. Second, whether the Bosonic mode can be explained as low energy phonon or have to be a Leggett mode. This is an important but difficult question from the experimental side. In this work, the mode energy smoothly varies as a function of doping, which might serve as a clue that the mode feature comes from the coupling of superconducting order parameters other than the lattice degree of freedom. Thus, the explanation of Leggett mode is at least self-consistent. Third, regarding to comment 5, I think it would be better to follow the Reviewer's suggestion. The plot of T_c as a function of mode energy has been discussed in cuprates and iron-based SCs, and it basically follow a line. For these low T_c samples, it is interesting to know that the ratio can deviate from the line, but its nature is not known yet. It is completely possible that the reason for the deviation can be different among kagome SCs, twisted graphenes, and 2D NbSe₂. Thus, the category of two regions is not very meaningful at this stage. The authors can delete the marks of two regions and just leave the data there.

Reviewer #3 (Remarks to the Author):

I would like to thank authors for responses. While the authors claim in Fig. R5_1 b and Fig. R5_2b that there are signatures of the Leggett mode around $E \sim \pm \Omega$ in the Josephson tunneling spectrum (that is mixed with the single particle tunneling spectrum), none of them are convincingly enough as the evidence of the Leggett mode. Overall, I don't find any evidence of the Leggett mode even in the Josephson tunneling in the revised manuscript. Unfortunately, I don't recommend for a publication in the Nature Communications. In the following, my comments are highlighted in purple.

Responses to the comments from the reviewers

Response to the Reviewer #3

I appreciate the authors for responses to my comments. I am also grateful to the authors for additional Josephson Tunneling measurements in the responses. While Fig. R1a captures the signature of the Josephson Tunneling, this looks just like a conventional Josephson Tunneling, and I don't see a clear signature associated with the Leggett mode in the data. Please see details below. I have to say that the manuscript lacks experimental evidence for the presence of the Leggett mode. I don't think that the peak-dip-hump structure in the local density of states can be evidence of the Leggett mode in the present study. Thus, I can't recommend for a publication in the Nature Communications. My responses are in green.

We thank the reviewer for continuing to provide constructive criticisms, comments, and suggestions. We have conducted further analyses and experiments to address the reviewer's concerns. We hope that these inputs will convince the reviewer that the peak-dip-hump structures is associated with Leggett mode, making it suitable for publication in Nature Communications.

In the following, for convenience of reviewer, reviewer's previous comments are in black italic typeface, our previous responses are in dark blue, the previous changes to the text are in gold.

In this round, our responses are in regular blue typeface and our changes to the text are in red.

Comment 1: The authors just analyzed ~ 300 dI/dV spectra for each x . First, it's not clear where these spectra are taken exactly in the field of view. Second and more importantly, it's not trivial that ~ 300 spectra are statistically enough to account for variations in the spectral shape (or heterogeneity in the electronic structure). It's possible that these spectra may not be representative but may be selected from local regions that is not an intrinsic property in this compound. Obviously, there are variations in the spectral shape as shown in Fig. 2 and 3, and the authors should demonstrate that ~ 300 spectra are enough. For example, one can take a dI/dV map in 50nm FOV with 256×256 pixels at given energy and construct a histogram of the values in dI/dV map. Then, one can simply crop/divide this map into four regions (4×25 nm 2128×128 pixels maps) and construct the histogram again for each map. If the size of the FOV and pixels# are statistically enough, then I would expect that widths of the distribution are the same before and after the crop.

Alternatively, and more convincingly, the authors can take a data in the same way as in the Ref. 2, constructing both Δ and Ω maps and visualizing spatial structure of them.

Response 1: We thank the reviewer for the suggestion on constructing $\Delta(\mathbf{r})$ and $\Omega(\mathbf{r})$ maps. Fig.R2 in the revised manuscript shows the result of $\Delta(\mathbf{r})$ and $\Omega(\mathbf{r})$ collected within a field of view of 25×25 nm and 180×180 pixels. The spatial variation and fertile patterns are visualized clearly, implying the spatial variation of dI/dV spectra in CsV3Sb5.

Fig. R2. Spatial distribution of $\Delta(\mathbf{r})$ and $\Omega(\mathbf{r})$ of CsV3Sb5. a, Spatially distribution of $\Delta(\mathbf{r})$ obtained on the Sb surface within the field of view of $25 \text{ nm} \times 25 \text{ nm}$, showing the spatially inhomogeneity of superconducting gap ($V_s = -2.0 \text{ mV}$, $I_t = 1 \text{ nA}$, $V_{\text{mod}} = 0.1 \text{ mV}$). b, The histogram plot of a. c, Spatial distribution of $\Omega(\mathbf{r})$ obtained on the same field of view of a, showing the spatial inhomogeneity of the bosonic mode. d, The histogram plot of c.

Accordingly, we added spatial distribution of $\Delta(\mathbf{r})$ and $\Omega(\mathbf{r})$ as new Extended Data Fig. 2 and revised the statement to clarify the methodology in the revised manuscript, as follows:

“In Fig. 2c, we show a series of dI/dV curves with the dominant one-gap line-shape in a stackplot without offset from different regions and samples. The spatial distribution of $\Delta(\mathbf{r})$ and $\Omega(\mathbf{r})$ within a field of view of $25 \text{ nm} \times 25 \text{ nm}$ are presented in Extended Data Fig. 2.”

New Comment 1: I understand the statistics. But it looks to me that there is a tip change in the data, and data quality at the top 25% of the FOV is quite different where the gap values are highly uniform around 0.35 meV . This would largely contribute to the histogram in Fig. R1b.

Nevertheless, a full width the half maximum is about 0.02 meV that is about 6% from its average. This suggests that the Δ is highly uniform compared to other compounds such as those in cuprates.

Response 1: We thank the reviewer for bringing up this question. As the reviewer has pointed out, it is important to check if the tip change in the $\Delta(\mathbf{r})$. The three aspects on topography $T(\mathbf{r})$, dI/dV spectra, and dI/dV maps are as follow:

Figure R2_1 a is the simultaneously obtained $T(\mathbf{r})$ of $\Delta(\mathbf{r})$, which shows the tip condition is preserved during the measurement.

Figure R2_1 b is the stack plot of dI/dV spectra with a step of 5 pixels as in $\Delta(\mathbf{r})$, which shows the superconducting gap varies smoothly within the statistical histogram during the measurement.

Figure R2_1 c-f are the typical dI/dV maps in energy range from 0.3 meV to 0.8 meV in line with the histogram in Fig. R2b, which show the preserved tip condition during the measurement.

Fig. R2_1. Topography, STS and dI/dV maps. a, Topography $T(\mathbf{r})$ obtained simultaneously and in the same field of view of $\Delta(\mathbf{r})$. b, A series of dI/dV spectra obtained simultaneously and in the same field of view of a, which generates the $\Delta(\mathbf{r})$. c-f, A series of dI/dV maps obtained at energy of -0.333 meV (c), -0.347 meV (d), -0.360 meV (e), -0.373 meV (f).

It also should be noticed that the simultaneously obtained $\Omega(\mathbf{r})$ in Fig. R2c do not show the signature regarding the tip change. Although the superconducting gap size varies in a small field of view, the inhomogeneity of the superconductor lays on the coherence peaks, in-gap conductance, and the Bogoliubov quasiparticle density of states beyond the superconducting gap, especially compared to the Ta-substituted components.

On the other hand, we conduct new $\Delta(\mathbf{r})$ map within a field view of 15 nm \times 15 nm as shown in Fig. R2_2 (new Supplementary Figure 2). The full width of the half maximum is about 0.02 meV as well.

Fig. R2_2 (new Supplementary Figure 2). Spatial distribution of $\Delta(\mathbf{r})$ and $\Omega(\mathbf{r})$ of

CsV3Sb5. a, Spatially distribution of $\Delta(\mathbf{r})$ obtained on the Sb surface within the field of view of 15 nm \times 15 nm, showing the spatially inhomogeneity of superconducting gap ($V_s = -2.0$ mV,

$I_t = 1$ nA, $V_{\text{mod}} = 0.1$ mV). b, The histogram plot of a. c, Spatial distribution of $\Omega(\mathbf{r})$ obtained on the same field of view of a, showing the spatial inhomogeneity of the bosonic mode. d, The histogram plot of c.

I would like to thank the authors for clarification. The only thing I am concerned about here is the word ‘inhomogeneous’. While the authors claim that the Δ is inhomogeneous, quantitatively it’s not true – if an inhomogeneity is estimated from the Full-Width at the Half Maximum in the distributions in Fig. R2_2c and d, then, “inhomogeneities” in Δ and Ω are $\sim 5\text{-}6\%$ from their average values. The cuprate superconductors are a prototypical inhomogeneous material, and it is estimated that the corresponding inhomogeneity for Δ is $\sim 50\%$ (Fig. 4g in J. W. Alldredge et al, Nature Phys. 4, 319 (2008)), and Ω inhomogeneity is $\sim 20\%$ (Fig. 4b in J. Lee et al, Nature 442, 546 (2006)). Δ inhomogeneity in the present system is roughly one order of magnitude more “homogeneous” than the cuprates. Thus, I think that ‘inhomogeneous’ is misleading. Δ is rather regarded as being homogeneous.

Comment 2: A Leggett mode energy is proportional to the superconducting order parameter amplitude (S.G. Sharapov et al., Eur. Phys. J. B 30, 45–51 (2002)). Thus, I would expect that the mode energy Ω should increase with increasing the gap size Δ , unless a coupling constant changes dramatically with x . However, as shown in Fig. 4, Ω and Δ show an opposite trend - Ω decreases but Δ increases with x . Why do they show the opposite trend with x ?

Response 2: We thank the reviewer for this critical comment. In the primary work studied by Professor A. J. Leggett (Prog. Theor. Phys. 36, 901-930 (1966)), the mode energy expressed as:

where $\rho_{1,2}$ is the density of states at Fermi surface, U and V are the pairing potential, J is the Josephson coupling between the two superconducting components, $\Delta_{1,2}$ is the superconducting

order parameter. It is natural to expect a positive correlation between the superconducting gap (or T_c) and mode energy when assuming an unchanged density of states.

However, with the introduction of Ta substitution, the normal state density will get an enhancement, stemming from the weakening of the charge density wave. When the change in the normal state density is much significant than in the superconducting gap, an anti-correlation between the superconducting gap and bosonic mode energy emerges.

It should be also noted that the anti-correlation between superconducting gap and Leggett mode energy has been well-studied in single-layer NbSe₂ (Adv. Mater. 34, e2206078 (2022)).

To avoid confusion on the anti-correlation between the superconducting gap and Leggett mode energy emerges in the Kagome superconductors, we have added discussions in the revised manuscript as follows:

“We next turn to discuss the possible origin of the bosonic mode. In Fig. 4c, the mode energy Ω is plotted as a function of the Ta-substitution x . It decreases continuously with increasing x accompanied by the increase of the SC gap Δ defined by the peak-to-peak distance in the conductance spectra and the suppression of the CDW order (Fig. 3). Such an anti-correlation between the Leggett mode and the superconducting gap has also been observed in monolayer NbSe₂. The Leggett mode is expected to be proportional to the SC gaps but anti-correlated with the density of states¹². When the CDW order in kagome superconductors is suppressed, the normal state density is likely to gain a more significant enhancement than the superconducting gap, resulting in an anti-correlation relationship.”

New Comment 2: I am not convinced by these arguments. In general, in the CDW ordered phase, the Fermi surface is expected to be reconstructed and be (partially) gapped out. ρ_1 and ρ_2 are the density of states at the Fermi level, so they are supposed to be (partially) gapped out in the CDW ordered phase. So, I don't understand why ρ_1 and ρ_2 are enhanced by weakening the CDW states. ρ_1 and ρ_2 are expected to be independent on the magnitude of the CDW order parameter.

Response 2: We thank the reviewer for the insights regarding the Fermi surface and the density of states in the presence of CDW. For both superconductivity and CDW, the density of states at the Fermi level plays an essential role (npj Quantum Materials 5, 22 (2020), Supercond. Sci. Technol. 14, R1–R27 (2001)). The reviewer correctly points out that the density of states at the Fermi level could be reduced in the CDW ordered phase. In turn, suppression of the CDW by

Ta substitution begins to close the CDW gap, leading to the release of the previously gapped Fermi level density of states. An increase in ρ_1 and ρ_2 would then be expected. Furthermore, our recent ARPES work report the substantially enhanced density of states at the Fermi level induced by the Ta doping (Supplementary Fig. S19 in Nat. Commun. 14, 3819 (2023))

If Δ and Ω are inhomogeneous (although it seems misleading), the authors could evaluate their relationship within a sample at fixed Ta concentration by constructing a 2D histogram between $\Delta(r)$ and $\Omega(r)$ images. If the equation above is relevant to the present case, then one would expect a positive correlation between Δ and Ω , thus the 2D histogram would show a positively sloped anisotropic distribution in the Δ and Ω space. Independently, the authors could simply calculate a cross-correlation coefficient between $\Delta(r)$ and $\Omega(r)$ images.

Comment 6: I would suggest the authors to perform Josephson Tunneling Microscopy measurement using the superconducting tip. I would expect a feature associated with the Leggett mode in the Josephson current spectra if exists.

Response 6: We thank the reviewer for the insightful suggestion. We have constructed the Josephson tunnelling spectroscopy and investigated the bosonic mode further. Fig. R5 shows that the signal of bosonic mode becomes more significant by using the Josephson tunnelling spectroscopy, and the bosonic mode accompanied by the superconducting gap and Josephson current disappears when $T > T_c$, demonstrating a superconducting collective mode.

In the revised manuscript, we add the description on fabrication of superconducting STM tip in the method part as follows:

“Fabrication of the superconducting CsV3Sb5 nanoflake tip. The tungsten STM tip was used to inject the clean surface of the sample more than 15 nm depth, holding for 5~10 s with the voltage 1~2 V, and withdraw to its original position. After that, the CsV3Sb5 nanoflake will stick to the apex of the tungsten tip. We usually obtained the stable CsV3Sb5 nanoflake tip after repeating the “injection process” for many times at several clean as-cleaved surface regions.”

Fig. R5. Temperature-dependent dI/dV (d^2I/dV^2) spectra obtained using the normal STM tip and the superconducting STM tip. a, A series of dI/dV spectra obtained by normal STM tip under different temperatures, showing the superconducting gaps and the bosonic mode gradually disappear with increasing temperature, and are invisible when electron temperature

$T_{\text{electron}} > 2.06$ K ($V_s = -2.0$ mV, $V_{\text{mod}} = 0.1$ mV, $I_t = 1$ nA). b, A series of derivative spectra (d^2I/dV^2) obtained from a. c, A series of dI/dV spectra obtained by superconducting STM tip under different temperatures, showing clearly the signal of Josephson peak, superconducting coherence peak, and bosonic mode, and those signals are weakening with increasing temperature, giving a phase transition temperature of $T_{\text{electron}} \sim 2.00$ K ($V_s = -2.0$ mV, $V_{\text{mod}} = 0.1$ mV, $I_t = 600$ nA). d, A series of derivative spectra obtained from c. Superconducting peaks, bosonic mode, and the Josephson peak in a, b, c, and d are labeled and marked by the dashed lines.

New Comment 6: I am grateful to the authors for the Josephson tunneling measurements. The authors made a superconducting tip by picking up a CsV3Sb5 flake from the sample such that the spectral signatures on the tip are virtually identical to those on the sample. In the Josephson tunneling, I expect to see two different signatures – one is a conventional Josephson tunneling (“intra-band” Josephson tunneling between the same bands in the sample and tip), and another would be associated with a Leggett mode (“inter-band” Josephson tunneling between the different bands in the sample and tip). Thus, I expect to see a conventional Josephson branch in the tunneling near the zero bias and an enhancement of the current at $|V_{\text{bias}}| > \Omega$ due to Leggett mode. These signatures would show up as peaks near the zero bias and $|V_{\text{bias}}| \sim \Omega$ in dI/dV , respectively. Contributions from a single particle tunneling will be on top of them. I also expect to see a sudden change in these signatures at the temperature where a smaller gap in one band is closed. At this temperature, a Josephson tunneling associated with the Leggett mode would disappear as well as

the “intra-band” Josephson for the smaller gap, so that the corresponding spectral signatures would disappear below T_c . In the data presented in Fig. R5, I don’t see clear evidence of the Josephson tunneling associated with the Leggett mode. The authors just observed the conventional Josephson current in Fig. R5.

Response 6: We thank the reviewer for the insightful comments regarding the Josephson tunneling measurements. The reviewer is correct, inter-band and intra-band Josephson tunneling are both expected when using the CsV3Sb5 nanoflake tip (Fig. R5_1a). The intraband Josephson tunneling is in analogy with the famous “Fujita conjecture” as well-studied in cuprate (Nature 580, 65-70 (2020)).

It should be noticed that the signal at energy around $|\Omega|$ could merge into the coherence peak shoulder, then leading to a visually weak feature. Despite this challenge in experiments, we can still observe the peaks $|V_{bias}| \sim \Omega$ in dI/dV when the “inter-band” Josephson tunneling between the different bands in the sample and tip is dominated. As shown in Fig. R5_1 (new Supplementary Figure 4), the symmetric peaks tailing to the coherence peaks are labeled.

Fig. R5_1 (new Supplementary Figure 4). Two types of Josephson tunnelling and the signature of Leggett mode. a, Schematic of “intra-band” Josephson tunneling between the same band of sample and tip, and “inter-band” Josephson tunneling between different bands of sample and tip. b, The dI/dV spectrum obtained by the superconducting STM tip, showing the bosonic mode manifest itself at energy $\sim \Omega$ ($V_s = -10.0$ mV, $V_{mod} = 0.1$ mV, $I_t = 1200$ nA).

While the authors forcefully highlight features at $E = \pm\Omega$ in Fig. R5_1b, I don’t see any features there. I don’t think this claim is acceptable.

We thank the reviewer for the great suggestion on temperature-dependent Josephson tunneling measurements. Let us focus on Fig. R5_2 at first. Since the peaks at energy around $|\Omega|$ are weak and could merged into the convolved coherence peak, we mark the dips and peaks in the derivative dI/dV spectra in Fig. R5_2, where the collective mode manifested at energies around $|\Omega|$ and $|\Delta + \Omega|$. With increasing temperature to ~ 1.86 K, the signal at energy around $|\Omega|$ disappears due the disappearance of the smaller gap as pointed out by the reviewer, while the signal at energy around $|\Delta + \Omega|$ can continue to be observed.

Fig. R5_2. Temperature-dependent dI/dV (d^2I/dV^2) spectra obtained using the superconducting STM tip. a, A series of dI/dV spectra obtained by superconducting STM tip under different temperatures, showing clearly the signal of Josephson peak, superconducting coherence peak, and bosonic mode, and those signals are weakening with increasing temperature, giving a phase transition temperature of Telectron ~ 2.00 K ($V_s = -2.0$ mV, $V_{mod} = 0.1$ mV, $I_t = 600$ nA). b, A series of derivative spectra obtained from a. The blue dashed line in a mark the superconducting peaks and the Josephson peak. The black dashed lines in b mark the evolution of Δ and $|\Delta + \Omega|$.

However, it is not feasible for us to perform a deeper investigation of the Leggett mode evolution associated with “intra-band” and “inter-band” Josephson tunneling in experiments, due to the resolution of our instrument and the challenge of keeping the temperature at around 2 K for a long time. On the other hand, the Leggett mode could couple to the Higgs mode as introduced in the manuscript, which could also weaken the spectral signal at $|E| = \Omega$.

Following the reviewer’s insightful suggestion, we can look at the pure Leggett in the feature. Accordingly, we add a new method part to revised manuscript:

Discussion on the Leggett mode in Josephson tunneling spectroscopy. CsV₃Sb₅ is a two-band superconductor, so two types of Josephson tunneling, namely “intra-band” and “inter-band” Josephson tunneling (Supplementary Fig. 4a), will contribute to the experimental signal when the CsV₃Sb₅ nanoflake tip is used. This scenario is in analogy to the well-known “Fujita conjecture” proposed for cuprates. As shown in Supplementary Fig. 4a, the “intra-band” Josephson tunneling is between the same band in the sample and the tip, and the “inter-band” Josephson tunneling is between different bands in the sample and the tip. In this scenario, the Leggett mode is expected to enhance the tunneling current when the energy is greater than Ω , which is manifested as a peak in the dI/dV spectrum. Supplementary Figure. 4b shows the signature of the Leggett mode at $|E| = \Omega$.

And the main text is revised according to Method:

In addition, the bosonic mode manifest itself at energy around Ω is also observed when using the superconducting STM tip, which is due to the potential “inter-band” Josephson tunneling (Method).

I am grateful to the authors for elaborating on this issue. While the authors highlighted in Fig. R5_2 b that there are features around $E \sim \pm\Omega$, these are identical to those the authors labeled as Δ in Fig. R5_1d. I don’t understand how the authors differentiate these two features. Please see an image below, in which I overlapped Fig. R5_1d and Fig. R5_2b together. These measurements are indeed impressive, but I don’t see any feature that is associated with the Leggett mode in the Josephson Tunneling. I don’t think I can accept claims the authors made here.

I would like to thank authors for responses. While the authors claim in Fig. R5_1 b and Fig. R5_2b that there are signatures of the Leggett mode around $E \sim \pm \Omega$ in the Josephson tunneling spectrum (that is mixed with the single particle tunneling spectrum), none of them are convincingly enough as the evidence of the Leggett mode. Overall, I don't find any evidence of the Leggett mode even in the Josephson tunneling in the revised manuscript. Unfortunately, I don't recommend for a publication in the Nature Communications. In the following, my comments are highlighted in purple.

Responses to the comments from the reviewers

Response to the Reviewer #3

I appreciate the authors for responses to my comments. I am also grateful to the authors for additional Josephson Tunneling measurements in the responses. While Fig. R1a captures the signature of the Josephson Tunneling, this looks just like a conventional Josephson Tunneling, and I don't see a clear signature associated with the Leggett mode in the data. Please see details below. I have to say that the manuscript lacks experimental evidence for the presence of the Leggett mode. I don't think that the peak-dip-hump structure in the local density of states can be evidence of the Leggett mode in the present study. Thus, I can't recommend for a publication in the Nature Communications. My responses are in green.

We thank the reviewer for continuing to provide constructive criticisms, comments, and suggestions. We have conducted further analyses and experiments to address the reviewer's concerns. We hope that these inputs will convince the reviewer that the peak-dip-hump structures is associated with Leggett mode, making it suitable for publication in Nature Communications.

In the following, for convenience of reviewer, reviewer's previous comments are in *black italic* typeface, our previous responses are in **dark blue**, the previous changes to the text are in **gold**. In this round, our responses are in **regular blue** typeface and our changes to the text are in **red**.

Comment 1: *The authors just analyzed ~300 dI/dV spectra for each x . First, it's not clear where these spectra are taken exactly in the field of view. Second and more importantly, it's not trivial that ~300 spectra are statistically enough to account for variations in the spectral shape (or heterogeneity in the electronic structure). It's possible that these spectra may not be representative but may be selected from local regions that is not an intrinsic property in this compound. Obviously, there are variations in the spectral shape as shown in Fig. 2 and 3, and the authors should demonstrate that ~300 spectra are enough. For example, one can take a dI/dV map in 50nm FOV with 256×256 pixels at given energy and construct a histogram of the*

values in dI/dV map. Then, one can simply crop/divide this map into four regions ($4 \times 25 \text{ nm}^2$ 128×128 pixels maps) and construct the histogram again for each map. If the size of the FOV and pixels# are statistically enough, then I would expect that widths of the distribution are the same before and after the crop. Alternatively, and more convincingly, the authors can take a data in the same way as in the Ref. 2, constructing both Δ and Ω maps and visualizing spatial structure of them.

Response 1: We thank the reviewer for the suggestion on constructing $\Delta(\mathbf{r})$ and $\Omega(\mathbf{r})$ maps. Fig.R2 in the revised manuscript shows the result of $\Delta(\mathbf{r})$ and $\Omega(\mathbf{r})$ collected within a field of view of $25 \times 25 \text{ nm}$ and 180×180 pixels. The spatial variation and fertile patterns are visualized clearly, implying the spatial variation of dI/dV spectra in CsV_3Sb_5 .

Fig. R2. Spatial distribution of $\Delta(\mathbf{r})$ and $\Omega(\mathbf{r})$ of CsV_3Sb_5 . **a**, Spatially distribution of $\Delta(\mathbf{r})$ obtained on the Sb surface within the field of view of $25 \text{ nm} \times 25 \text{ nm}$, showing the spatially inhomogeneity of superconducting gap ($V_s = -2.0 \text{ mV}$, $I_f = 1 \text{ nA}$, $V_{mod} = 0.1 \text{ mV}$). **b**, The histogram plot of **a**. **c**, Spatial distribution of $\Omega(\mathbf{r})$ obtained on the same field of view of **a**, showing the spatial inhomogeneity of the bosonic mode. **d**, The histogram plot of **c**.

Accordingly, we added spatial distribution of $\Delta(\mathbf{r})$ and $\Omega(\mathbf{r})$ as new Extended Data Fig. 2 and revised the statement to clarify the methodology in the revised manuscript, as follows:

“In Fig. 2c, we show a series of dI/dV curves with the dominant one-gap line-shape in a stackplot without offset from different regions and samples. The spatial distribution of $\Delta(\mathbf{r})$ and $\Omega(\mathbf{r})$ within a field of view of $25 \text{ nm} \times 25 \text{ nm}$ are presented in Extended Data Fig. 2.”

New Comment 1: I understand the statistics. But it looks to me that there is a tip change in the data, and data quality at the top 25% of the FOV is quite different where the gap values are highly uniform around 0.35meV. This would largely contribute to the histogram in Fig. R1b. Nevertheless, a full width the half maximum is about 0.02meV that is about 6% from its average. This suggests that the Δ is highly uniform compared to other compounds such as those in cuprates.

Response 1: We thank the reviewer for bringing up this question. As the reviewer has pointed out, it is important to check if the tip change in the $\Delta(\mathbf{r})$. The three aspects on topography $T(\mathbf{r})$, dI/dV spectra, and dI/dV maps are as follow:

1. Figure R2_1 a is the simultaneously obtained $T(\mathbf{r})$ of $\Delta(\mathbf{r})$, which shows the tip condition is preserved during the measurement.
2. Figure R2_1 b is the stack plot of dI/dV spectra with a step of 5 pixels as in $\Delta(\mathbf{r})$, which shows the superconducting gap varies smoothly within the statistical histogram during the measurement.
3. Figure R2_1 c-f are the typical dI/dV maps in energy range from 0.3 meV to 0.8 meV in line with the histogram in Fig. R2b, which show the preserved tip condition during the measurement.

Fig. R2_1. Topography, STS and dI/dV maps. **a**, Topography $T(r)$ obtained simultaneously and in the same field of view of $\Delta(r)$. **b**, A series of dI/dV spectra obtained simultaneously and in the same field of view of **a**, which generates the $\Delta(r)$. **c-f**, A series of dI/dV maps obtained at energy of -0.333 meV (**c**), -0.347 meV (**d**), -0.360 meV (**e**), -0.373 meV (**f**).

It also should be noticed that the simultaneously obtained $\Omega(r)$ in Fig. R2c do not show the signature regarding the tip change. Although the superconducting gap size varies in a small field of view, the inhomogeneity of the superconductor lays on the coherence peaks, in-gap conductance, and the Bogoliubov quasiparticle density of states beyond the superconducting gap, especially compared to the Ta-substituted components.

On the other hand, we conduct new $\Delta(r)$ map within a field view of $15 \text{ nm} \times 15 \text{ nm}$ as shown in Fig. R2_2 (new Supplementary Figure 2). The full width of the half maximum is about 0.02 meV as well.

Fig. R2_2 (new Supplementary Figure 2). Spatial distribution of $\Delta(r)$ and $\Omega(r)$ of CsV_3Sb_5 . **a**, Spatially distribution of $\Delta(r)$ obtained on the Sb surface within the field of view of $15 \text{ nm} \times 15 \text{ nm}$, showing the spatial inhomogeneity of superconducting gap ($V_s = -2.0 \text{ mV}$, $I_T = 1 \text{ nA}$, $V_{mod} = 0.1 \text{ mV}$). **b**, The histogram plot of **a**. **c**, Spatial distribution of $\Omega(r)$ obtained on the same field of view of **a**, showing the spatial inhomogeneity of the bosonic mode. **d**, The histogram plot of **c**.

I would like to thank the authors for clarification. The only thing I am concerned about here is the word ‘inhomogeneous’. While the authors claim that the Δ is inhomogeneous, quantitatively it’s not true – if an inhomogeneity is estimated from the Full-Width at the Half Maximum in the distributions in Fig. R2_2c and d, then, “inhomogeneities” in Δ and Ω are ~5-6% from their average values. The cuprate superconductors are a prototypical inhomogeneous material, and it is estimated that the corresponding inhomogeneity for Δ is ~50% (Fig. 4g in J. W. Allredge *et al*, *Nature Phys.* **4**, 319 (2008)), and Ω inhomogeneity is ~20% (Fig. 4b in J. Lee *et al*, *Nature* **442**, 546 (2006)). Δ inhomogeneity in the present system is roughly one order of magnitude more “homogeneous” than the cuprates. Thus, I think that ‘inhomogeneous’ is misleading. Δ is rather regarded as being homogeneous.

Comment 2: *A Leggett mode energy is proportional to the superconducting order parameter amplitude (S.G. Sharapov *et al.*, *Eur. Phys. J. B* **30**, 45–51 (2002). Thus, I would expect that the mode energy Ω should increase with increasing the gap size Δ , unless a coupling constant changes dramatically with x . However, as shown in Fig. 4, Ω and Δ show an opposite trend - Ω decreases but Δ increases with x . Why do they show the opposite trend with x ?*

Response 2: We thank the reviewer for this critical comment. In the primary work studied by Professor A. J. Leggett (*Prog. Theor. Phys.* **36**, 901-930 (1966)), the mode energy expressed as:

$$E \propto 4 \sqrt{\frac{1}{2} \left(\frac{1}{\rho_1} + \frac{1}{\rho_2} \right) \left(\frac{J}{UV} \right) \Delta_1 \Delta_2}$$

where $\rho_{1,2}$ is the density of states at Fermi surface, U and V are the pairing potential, J is the Josephson coupling between the two superconducting components, $\Delta_{1,2}$ is the superconducting order parameter. It is natural to expect a positive correlation between the superconducting gap (or T_c) and mode energy when assuming an unchanged density of states.

However, with the introduction of Ta substitution, the normal state density will get an enhancement, stemming from the weakening of the charge density wave. When the change in the normal state density is much significant than in the superconducting gap, an anti-correlation between the superconducting gap and bosonic mode energy emerges.

It should be also noted that the anti-correlation between superconducting gap and Leggett mode energy has been well-studied in single-layer NbSe₂ (*Adv. Mater.* **34**, e2206078 (2022)).

To avoid confusion on the anti-correlation between the superconducting gap and Leggett mode energy emerges in the Kagome superconductors, we have added discussions in the revised manuscript as follows:

“We next turn to discuss the possible origin of the bosonic mode. In Fig. 4c, the mode energy Ω is plotted as a function of the Ta-substitution x . It decreases continuously with increasing x accompanied by the increase of the SC gap Δ defined by the peak-to-peak distance in the conductance spectra and the suppression of the CDW order (Fig. 3). Such an anti-correlation between the Leggett mode and the superconducting gap has also been observed in monolayer NbSe₂¹⁴. The Leggett mode is expected to be proportional to the SC gaps but anti-correlated with the density of states¹². When the CDW order in kagome superconductors is suppressed, the normal state density is likely to gain a more significant enhancement than the superconducting gap, resulting in an anti-correlation relationship.”

New Comment 2: I am not convinced by these arguments. In general, in the CDW ordered phase, the Fermi surface is expected to be reconstructed and be (partially) gapped out. ρ_1 and ρ_2 are the density of states at the Fermi level, so they are supposed to be (partially) gapped out in the CDW ordered phase. So, I don't understand why ρ_1 and ρ_2 are enhanced by weakening the CDW states. ρ_1 and ρ_2 are expected to be independent on the magnitude of the CDW order parameter.

Response 2: We thank the reviewer for the insights regarding the Fermi surface and the density of states in the presence of CDW. For both superconductivity and CDW, the density of states at the Fermi level plays an essential role (*npj Quantum Materials* **5**, 22 (2020), *Supercond. Sci. Technol.* **14**, R1–R27 (2001)). The reviewer correctly points out that the density of states at the Fermi level could be reduced in the CDW ordered phase. In turn, suppression of the CDW by Ta substitution begins to close the CDW gap, leading to the release of the previously gapped Fermi level density of states. An increase in ρ_1 and ρ_2 would then be expected. Furthermore, our recent ARPES work report the substantially enhanced density of states at the Fermi level induced by the Ta doping (Supplementary Fig. S19 in *Nat. Commun.* **14**, 3819 (2023))

If Δ and Ω are inhomogeneous (although it seems misleading), the authors could evaluate their relationship within a sample at fixed Ta concentration by constructing a 2D histogram between $\Delta(r)$ and $\Omega(r)$ images. If the equation above is relevant to the present case, then one would expect a positive correlation between Δ and Ω , thus the 2D histogram would show a positively sloped anisotropic distribution in the Δ and Ω space. Independently, the authors could simply calculate a cross-correlation coefficient between $\Delta(r)$ and $\Omega(r)$ images.

Comment 6: I would suggest the authors to perform Josephson Tunneling Microscopy measurement using the superconducting tip. I would expect a feature associated with the Leggett mode in the Josephson current spectra if exists.

Response 6: We thank the reviewer for the insightful suggestion. We have constructed the Josephson tunnelling spectroscopy and investigated the bosonic mode further. Fig. R5 shows that the signal of bosonic mode becomes more significant by using the Josephson tunnelling spectroscopy, and the bosonic mode accompanied by the superconducting gap and Josephson current disappears when $T > T_c$, demonstrating a superconducting collective mode.

In the revised manuscript, we add the description on fabrication of superconducting STM tip in the method part as follows:

“Fabrication of the superconducting CsV₃Sb₅ nanoflake tip. The tungsten STM tip was used to inject the clean surface of the sample more than 15 nm depth, holding for 5~10 s with the voltage 1~2 V, and withdraw to its original position. After that, the CsV₃Sb₅ nanoflake will stick to the apex of the tungsten tip. We usually obtained the stable CsV₃Sb₅ nanoflake tip after repeating the “injection process” for many times at several clean as-cleaved surface regions.”

Fig. R5. Temperature-dependent dI/dV (d^2I/dV^2) spectra obtained using the normal STM tip and the superconducting STM tip. a, A series of dI/dV spectra obtained by normal STM tip under different temperatures, showing the superconducting gaps and the bosonic mode gradually disappear with increasing temperature, and are invisible when electron temperature

$T_{\text{electron}} > 2.06$ K ($V_s = -2.0$ mV, $V_{\text{mod}} = 0.1$ mV, $I_t = 1$ nA). **b**, A series of derivative spectra (d^2I/dV^2) obtained from **a**. **c**, A series of dI/dV spectra obtained by superconducting STM tip under different temperatures, showing clearly the signal of Josephson peak, superconducting coherence peak, and bosonic mode, and those signals are weakening with increasing temperature, giving a phase transition temperature of $T_{\text{electron}} \sim 2.00$ K ($V_s = -2.0$ mV, $V_{\text{mod}} = 0.1$ mV, $I_t = 600$ nA). **d**, A series of derivative spectra obtained from **c**. Superconducting peaks, bosonic mode, and the Josephson peak in **a**, **b**, **c**, and **d** are labeled and marked by the dashed lines.

New Comment 6: I am grateful to the authors for the Josephson tunneling measurements. The authors made a superconducting tip by picking up a CsV_3Sb_5 flake from the sample such that the spectral signatures on the tip are virtually identical to those on the sample. In the Josephson tunneling, I expect to see two different signatures – one is a conventional Josephson tunneling (“intra-band” Josephson tunneling between the same bands in the sample and tip), and another would be associated with a Leggett mode (“inter-band” Josephson tunneling between the different bands in the sample and tip). Thus, I expect to see a conventional Josephson branch in the tunneling near the zero bias and an enhancement of the current at $|V_{\text{bias}}| > \Omega$ due to Leggett mode. These signatures would show up as peaks near the zero bias and $|V_{\text{bias}}| \sim \Omega$ in dI/dV , respectively. Contributions from a single particle tunneling will be on top of them. I also expect to see a sudden change in these signatures at the temperature where a smaller gap in one band is closed. At this temperature, a Josephson tunneling associated with the Leggett mode would disappear as well as the “intra-band” Josephson for the smaller gap, so that the corresponding spectral signatures would disappear below T_c . In the data presented in Fig. R5, I don’t see clear evidence of the Josephson tunneling associated with the Leggett mode. The authors just observed the conventional Josephson current in Fig. R5.

Response 6: We thank the reviewer for the insightful comments regarding the Josephson tunneling measurements. The reviewer is correct, inter-band and intra-band Josephson tunneling are both expected when using the CsV_3Sb_5 nanoflake tip (Fig. R5_1a). The intraband Josephson tunneling is in analogy with the famous “Fujita conjecture” as well-studied in cuprate (*Nature* **580**, 65-70 (2020)).

It should be noticed that the signal at energy around $|\Omega|$ could merge into the coherence peak shoulder, then leading to a visually weak feature. Despite this challenge in experiments, we can still observe the peaks $|V_{\text{bias}}| \sim \Omega$ in dI/dV when the “inter-band” Josephson tunneling between the different bands in the sample and tip is dominated. As shown in Fig. R5_1 (new Supplementary Figure 4), the symmetric peaks tailing to the coherence peaks are labeled.

Fig. R5_1 (new Supplementary Figure 4). Two types of Josephson tunnelling and the signature of Leggett mode. **a**, Schematic of “intra-band” Josephson tunneling between the same band of sample and tip, and “inter-band” Josephson tunneling between different bands of sample and tip. **b**, The dI/dV spectrum obtained by the superconducting STM tip, showing the bosonic mode manifest itself at energy $\sim \Omega$ ($V_s = -10.0$ mV, $V_{mod} = 0.1$ mV, $I_t = 1200$ nA).

While the authors forcefully highlight features at $E = \pm\Omega$ in Fig. R5_1b, I don't see any features there. I don't think this claim is acceptable.

We thank the reviewer for the great suggestion on temperature-dependent Josephson tunneling measurements. Let us focus on Fig. R5_2 at first. Since the peaks at energy around $|\Omega|$ are weak and could merged into the convolved coherence peak, we mark the dips and peaks in the derivative dI/dV spectra in Fig. R5_2, where the collective mode manifested at energies around $|\Omega|$ and $|\Delta + \Omega|$. With increasing temperature to ~ 1.86 K, the signal at energy around $|\Omega|$ disappears due to the disappearance of the smaller gap as pointed out by the reviewer, while the signal at energy around $|\Delta + \Omega|$ can continue to be observed.

Fig. R5_2. Temperature-dependent dI/dV (d^2I/dV^2) spectra obtained using the superconducting STM tip. **a**, A series of dI/dV spectra obtained by superconducting STM tip under different temperatures, showing clearly the signal of Josephson peak, superconducting coherence peak, and bosonic mode, and those signals are weakening with increasing temperature, giving a phase transition temperature of $T_{\text{electron}} \sim 2.00$ K ($V_s = -2.0$ mV, $V_{\text{mod}} = 0.1$ mV, $I_t = 600$ nA). **b**, A series of derivative spectra obtained from **a**. The blue dashed line in **a** mark the superconducting peaks and the Josephson peak. The black dashed lines in **b** mark the evolution of Δ and $|\Delta + \Omega|$.

However, it is not feasible for us to perform a deeper investigation of the Leggett mode evolution associated with “intra-band” and “inter-band” Josephson tunneling in experiments, due to the resolution of our instrument and the challenge of keeping the temperature at around 2 K for a long time. On the other hand, the Leggett mode could couple to the Higgs mode as introduced in the manuscript, which could also weaken the spectral signal at $|E| = \Omega$.

Following the reviewer’s insightful suggestion, we can look at the pure Leggett in the feature.

Accordingly, we add a new method part to revised manuscript:

Discussion on the Leggett mode in Josephson tunneling spectroscopy. CsV_3Sb_5 is a two-band superconductor, so two types of Josephson tunneling, namely “intra-band” and “inter-band” Josephson tunneling (Supplementary Fig. 4a), will contribute to the experimental signal when the CsV_3Sb_5 nanoflake tip is used. This scenario is in analogy to the well-known “Fujita conjecture” proposed for cuprates. As shown in Supplementary Fig. 4a, the “intra-band” Josephson tunneling is between the same band in the sample and the tip, and the “inter-band” Josephson tunneling is between different bands in the sample and the tip. In this scenario, the Leggett mode is expected to enhance the tunneling current when the energy is greater than Ω , which is manifested as a peak in the dI/dV spectrum. Supplementary Figure. 4b shows the signature of the Leggett mode at $|E| = \Omega$.

And the main text is revised according to Method:

In addition, the bosonic mode manifest itself at energy around Ω is also observed when using the superconducting STM tip, which is due to the potential “inter-band” Josephson tunneling (Method).

I am grateful to the authors for elaborating on this issue. While the authors highlighted in Fig. R5_2 b that there are features around $E \sim \pm\Omega$, these are identical to those the authors labeled as Δ in Fig. R5_1d. I don’t understand how the authors differentiate these two features. Please see an image below, in which I overlapped Fig. R5_1d and Fig. R5_2b together. These measurements are indeed impressive, but I don’t see any feature that is associated with the Leggett mode in the Josephson Tunneling. I don’t think I can accept claims the authors made here.

Responses to the comments from the reviewers

Response to the Reviewer #1

In the response to the second round of the reviewing process, the authors have addressed the comments of the reviewers in a fairly satisfactory manner. They added the data for the temperature dependence of the bosonic mode energy in Supplementary Fig. 3d, supporting the Leggett mode scenario (the phonon origin could be excluded). While it is not clear whether the observed signal comes from the intra-band or inter-band Josephson tunnelling, as pointed out by reviewer #3, at least the overall data presented in the paper seem to be consistent with the Leggett mode (inter-band Josephson mode). The ultimate identification may be open to future works, but the presented data are already interesting and worth to be published. I recommend to add a discussion on this point (the intra-band or inter-band mode) not only in the method section but also in the main text. With this, I can support the publication in Nature Communications.

Response: We sincerely thank the reviewer for positive remarks. According to the reviewer's suggestion, we have added a new a discussion on intra-band and inter-band mode in the main text:

“For the CsV₃Sb₅ sample – insulator – CsV₃Sb₅ nanoflake tunneling junction, the “intra-band” Josephson tunneling between same band and the “inter-band” Josephson tunneling between different bands are all allowed. In addition, the bosonic mode manifest itself at energy around Ω is also observed when using the superconducting STM tip (Method).”

Response to the Reviewer #2

In the rebuttal letter, the authors response to the Reviewer's comments in details. There are basically three concerns by Reviewer #3. The first one is on the statistics of SC gap and Bosonic energy. In the letter, the authors have included more dI/dV maps and a new data set to support the claim; I find the experimental data being convincing to extract the gap energy and Bosonic energy. Second, whether the Bosonic mode can be explained as low energy phonon or have to be a Leggett mode. This is an important but difficult question from the experimental side. In this work, the mode energy smoothly varies as a function of doping, which might serve as a clue that the mode feature comes from the coupling of superconducting order parameters other than the lattice degree of freedom. Thus, the explanation of Leggett mode is at least self-consistent. Third, regarding to comment 5, I think it would be better to follow the Reviewer's suggestion. The plot of T_c as a function of mode energy has been discussed in cuprates and iron-based SCs, and it basically follows a line. For these low T_c samples, it is interesting to know that the ratio can deviate from the line, but its nature is not known yet. It is completely possible that the reason for the deviation can be different among kagome SCs, twisted graphenes, and 2D $NbSe_2$. Thus, the category of two regions is not very meaningful at this stage. The authors can delete the marks of two regions and just leave the data there.

Response 1: We thank the reviewer for the nice summary on the reviewer 3#'s concerns and positive remarks on our previous response. According to reviewer's suggestion, we have deleted the marks of two regions and just leave the data in the revised Fig. 4f.

Revised Fig. 4f. Plot of transition temperature T_c as a function of mode energy Ω . The data are taken or extracted from mostly STM works on kagome superconductors reported here, $NbSe_2$ ¹⁴, twisted bilayer graphene⁶¹, twisted trilayer graphene⁶², $Pr_{0.88}LaCe_{0.12}CuO_4$ ³, one-unit-cell FeSe⁴, $EuRbFe_4As_4$ ⁶³, $Ba_{0.6}K_{0.4}Fe_2As_2$ ^{50,51}, and $Na(Fe_{0.975}Co_{0.025})As$ ⁵⁰, as well as from the Uemura plot⁵. The T_c s of twisted trilayer graphene are extracted from Ref⁶⁴.

Response to the Reviewer #3

Comment 1: *I would like to thank the authors for clarification. The only thing I am concerned about here is the word “inhomogeneous”. While the authors claim that the Δ is inhomogeneous, quantitatively it’s not true – if an inhomogeneity is estimated from the Full-Width at the Half Maximum in the distributions in Fig. R2_2c and d, then, “inhomogeneities” in Δ and Ω are ~5- 6% from their average values. The cuprate superconductors are a prototypical inhomogeneous material, and it is estimated that the corresponding inhomogeneity for Δ is ~50% (Fig. 4g in J. W. Alldredge et al, Nature Phys. 4, 319 (2008)), and Ω inhomogeneity is ~20% (Fig. 4b in J. Lee et al, Nature 442, 546 (2006)). Δ inhomogeneity in the present system is roughly one order of magnitude more “homogeneous” than the cuprates. Thus, I think that ‘inhomogeneous’ is misleading. Δ is rather regarded as being homogeneous.*

Response 1: We thank the reviewer for the comment. We use word “inhomogeneity” solely to describe the spatial nonuniformity of the coherence peaks, in-gap conductance, and the Bogoliubov quasiparticle density of states beyond the superconducting gap, especially compared to the Ta-substituted components. We did not intend making an impression on that the CsV₃Sb₅ is prototypical inhomogeneous material and comparable to the cuprate where the Reviewer may misunderstand. To avoid the misleading, we have changed the “inhomogeneity” to “nonuniformity” throughout the manuscript.

Comment 2: *If Δ and Ω are inhomogeneous (although it seems misleading), the authors could evaluate their relationship within a sample at fixed Ta concentration by constructing a 2d histogram between $\Delta(r)$ and $\Omega(r)$ images. If the equation above is relevant to the present case, then one would expect a positive correlation between Δ and Ω , thus the 2D histogram would show a positively sloped anisotropic distribution in the Δ and Ω space. Independently, the authors could simply calculate a cross-correlation coefficient between $\Delta(r)$ and $\Omega(r)$ images.*

Response 2: We thank the reviewer for the critical comment. We plot the 2d histogram between $\Delta(r)$ and $\Omega(r)$ images extracted from the Fig. R2_2 of previous Response Letter (now Fig. R3). The $\Delta(r)$ and $\Omega(r)$ still shows the anti-correlation with a cross-correlation coefficient of -0.13.

Fig. R3. The 2d histogram between $\Delta(r)$ and $\Omega(r)$ images extracted from Fig. R2_2, where the calculated cross-correlation coefficient is -0.13, showing the anti-correlation between $\Delta(r)$ and $\Omega(r)$.

Comment 3: While the authors forcefully highlight features at $E = \pm\Omega$ in Fig. R4_1b, I don't see any features there. I don't think this claim is acceptable.

Response 3: We thank the reviewer for the comment. In our experiments, the conventional Josephson tunnelling with an enhancement of the current at $|V_{bias}| > \Omega$, which manifested as a peak in the dI/dV spectrum, is visually weak due to the merged signal of bosonic mode and SC coherence peak.

Comment 4: I am grateful to the authors for elaborating on this issue. While the authors highlighted in Fig. R4_2 b that there are features around $E \sim \pm\Omega$, these are identical to those the authors labeled as Δ in Fig. R4_1d. I don't understand how the authors differentiate these two features. Please see an image below, in which I overlapped Fig. R4_1d and Fig. R4_2b together. These measurements are indeed impressive, but I don't see any feature that is associated with the Leggett mode in the Josephson Tunneling. I don't think I can accept claims the authors made here.

Response 4: We thank the reviewer for the comment. In Fig. R4, the signal at $|V_{bias}| \sim \Omega$ is relatively weak and could merge into, thus coincident with the position of Δ . We mark the position of Ω from the light dip feature, while the superconducting gap Δ directly corresponds to the zero value in d^2I/dV^2 , which can be distinguished from the spectra.